# Cryo-EM reveals the architecture of the PELP1-WDR18 molecular scaffold

Jacob Gordon [1,2,3,4], Fleur L. Chapus[5], Elizabeth G. Viverette[6], Jason G. Williams[5], Leesa J. Deterding[5], Juno M. Krahn[6], Mario J. Borgnia [6], Joseph Rodriguez[5], Alan J. Warren [2,3,4] & Robin E. Stanley [1] ✉

PELP1 (Proline-, Glutamic acid-, Leucine-rich protein 1) is a large scaffolding protein that functions in many cellular pathways including steroid receptor (SR) coactivation, heterochromatin maintenance, and ribosome biogenesis. PELP1 is a proto-oncogene whose expression is upregulated in many human cancers, but how the PELP1 scaffold coordinates its diverse cellular functions is poorly understood. Here we show that PELP1 serves as the central scaffold for the human Rix1 complex whose members include WDR18, TEX10, and SENP3. We reconstitute the mammalian Rix1 complex and identified a stable sub-complex comprised of the conserved PELP1 Rix1 domain and WDR18. We determine a 2.7 Å cryo-EM structure of the subcomplex revealing an inter-connected tetrameric assembly and the architecture of PELP1's signaling motifs, including eleven LxxLL motifs previously implicated in SR signaling and coactivation of Estrogen Receptor alpha (ERα) mediated transcription. However, the structure shows that none of these motifs is in a conformation that would support SR binding. Together this work establishes that PELP1 scaffolds the Rix1 complex, and association with WDR18 may direct PELP1's activity away from SR coactivation.

Scaffolding proteins mediate many cellular processes by bringing together multiple binding partners to facilitate protein-protein inter-actions, enzymatic cascades, and intricate signaling pathways[1]. A characteristic feature of scaffolding proteins is the presence of multi-ple domains and signaling motifs. PELP1 is a well-known scaffolding protein with a unique amino-acid composition that has been impli-cated in numerous cellular activities[2,3]. Mammalian PELP1 is composed of two domains, including a well-conserved N-terminal domain refer-red to as the Rix1 domain (based on homology with the yeast

homologue Rix1), and a poorly conserved proline- and glutamic acid-rich C-terminal domain. The Rix1 domain of PELP1 encodes for numerous signaling motifs including eleven LxxLL and three PxxP (x is any amino acid) motifs that are both known for mediating protein-protein interactions during steroid receptor (SR) signaling[2-7]. PELP1 was originally discovered as a transcriptional coactivator of estrogen receptor alpha (ERα) and has since been shown to coregulate several major steroid receptors (SRs), including progesterone, androgen, and glucocorticoid receptors[4,8-12]. In addition to SRs, the PELP1 scaffold has

[1]Signal Transduction Laboratory, National Institute of Environmental Health Sciences, National Institutes of Health, Department of Health and Human Services, 111 T. W. Alexander Drive, Research Triangle Park, NC 27709, USA. [2]Cambridge Institute for Medical Research, Cambridge Biomedical Campus, Keith Peters Building, Hills Rd, Cambridge CB2 0XY, UK. [3]Wellcome Trust-Medical Research Council Stem Cell Institute, Jeffrey Cheah Biomedical Centre, Puddicombe Way, Cambridge Biomedical Campus, Cambridge CB2 0AW, UK. [4]Department of Haematology, University of Cambridge School of Clinical Medicine, Jeffrey Cheah Biomedical Centre, Puddicombe Way, Cambridge Biomedical Campus, Cambridge CB2 0AW, UK. [5]Epigenetics and Stem Cell Biology Laboratory, National Institute of Environmental Health Sciences, National Institutes of Health, Department of Health and Human Services, 111 T. W. Alexander Drive, Research Triangle Park, NC 27709, USA. [6]Genome Integrity and Structural Biology Laboratory, National Institute of Environmental Health Sciences, National Institutes of Health, Department of Health and Human Services, 111 T. W. Alexander Drive, Research Triangle Park, NC 27709, USA. ✉e-mail: robin.stanley@nih.gov

also been shown to support other transcription factors, mediators of the cell cycle, and chromatin-modifying enzymes[2,3,13].

Beyond supporting signaling networks, PELP1 plays fundamental cellular roles in ribosome assembly and maintenance of heterochromatin by serving as the central scaffold of a conserved eukaryotic multiprotein assembly known as the Rix1 complex[14–18]. The Rix1 complex is a large and essential molecular assembly containing the conserved PELP1-WDR18-TEX10 protein components (RIX1-IPI3-IPI1 in *S. cerevisiae*), with the inclusion of the SUMO protease SENP3 in mammals. Other than SENP3's SUMO-specific protease activity, the Rix1 complex has no other known enzymatic function. Previous reports have established that the Rix1 complex also associates with a multienzyme RNA processing complex called RNase PNK, composed of the endoribonuclease Las1 and the polynucleotide kinase Nol9[14,19,20]. Together, RNase PNK and the Rix1 complex form a conserved superassembly called the Rixosome[14,18]. The enzymatic activity of the Rixosome facilitates the processing of the preribosomal RNA and degradation of RNA from heterochromatin[14,18,21]. A lack of structures for any of the components of the human Rixosome has hindered our understanding of how the PELP1 scaffold supports Rixosome formation and function.

PELP1 is classified as a proto-oncogene and is dysregulated in many human hormonal cancers[2]. Previous work has established that 60-80% of breast cancer cases exhibit abnormal PELP1 expression, which is a predictor of poorer patient outcomes[2,3,22]. PELP1 dysregulation is also implicated in resistance to hormone therapy in breast cancers that are positive for ER expression[2,23–25]. Further, PELP1 can promote triple-negative breast cancer (TNBC)[26,27]. The strong association between PELP1 and many human cancers has established PELP1 as a promising therapeutic target, however little is known about how the PELP1 scaffold coordinates its diverse cellular roles.

In this work, to begin to decipher the structure and function of the PELP1 molecular scaffold, we sought to reconstitute and purify the human Rix1 complex for structural and biochemical characterization. Upon reconstitution of the human Rix1 complex, we discover the existence of a stable sub-complex comprised of the conserved Rix1 domain of PELP1 and WDR18. Using Cryo-electron microscopy (cryo-EM), we determine a 2.7 Å resolution structure of the human PELP1 Rix1 domain bound to WDR18. We visualize all eleven LxxLL motifs in PELP1's Rix1 domain as ordered α-helices, allowing for structure-informed reasoning of each LxxLL motifs' accessibility and SR-binding ability. We also investigate the function of PELP1-WDR18 in ERα coactivation through estrogen response element luciferase assays. We observe that WDR18 overexpressed with PELP1 exhibited reduced ERα-mediated transcription coactivation when compared to PELP1 alone. These experiments support a model in which PELP1-WDR18 assembly removes PELP1 from the pool of ERα coactivators within cancer cells, thus decreasing ERα coactivation.

## Results

### Reconstitution of the mammalian Rix1 complex

We reconstituted and purified the human Rix1 complex and characterized the molecular determinants of complex assembly. Previous work established PELP1 as the core stabilizing component for the human Rix1 complex[17]. However, little was known about the biochemical nature of the entire scaffolding assembly (Fig. 1a). We reconstituted the four-subunit human Rix1 complex (PELP1, WDR18, TEX10, and SENP3) with a mammalian suspension cell protein expression system. We assembled the human Rix1 complex by sequential transient expression of plasmids coding for epitope-tagged complex members in separate expression cultures. N-terminal FLAG-tagged PELP1 was used as bait for FLAG tag immunoprecipitation (IP) of the sequentially reconstituted complexes. FLAG IP complexes were analyzed by SDS-PAGE and Western blot for epitope-tagged complex members. Additionally, these complexes were eluted from the anti-

FLAG affinity gel in a native buffer and subjected to SDS-PAGE and total protein staining. Western blotting of sequentially built Rix1 complex members identified the association of each epitope-tagged member with PELP1 upon IP (Fig. 1b). Eluted complexes visualized by Coomassie protein staining exhibited similar results to that of Western blotting. Notably, the reconstitution of the conserved Rix1 complex members (PELP1-WDR18-TEX10) exhibited a weaker band signal for TEX10 compared to when SENP3 is present in the full complex (Fig. 1b, lane 4 vs. 5). These results suggest that SENP3 is important for the stable association of TEX10 within the Rix1 complex.

### Identification of the human PELP1-WDR18 sub-complex

Our reconstitution experiments led to the discovery of a stable PELP1-WDR18 subcomplex. An unexpected observation from our reconstitution experiments was the presence of a prominent second band at a molecular weight of approximately 70 kDa in addition to FLAG-PELP1 (180 kDa) in the anti-FLAG Western blot (Fig. 1b). This additional band was also observed in the Coomassie-stained gel of eluates. Considering the enrichment of the complexes using the N-terminal FLAG tag on PELP1, and the Western detection of the band using a FLAG antibody, we reasoned that this additional 70 kDa band is a fragment of the N-terminus of PELP1 in which the C-terminus has degraded. Mass spectrometry (MS) analysis of both 180 kDa and 70 kDa PELP1 bands revealed the absence of PELP1 C-terminal peptide spectra in the smaller 70 kDa species compared to the full-length 180 kDa species (data not shown). A similar observation was made in a previous report studying PELP1 overexpression[28].

The 1130 amino acids of PELP1 have a unique amino acid composition. Most notable is PELP1's acidic (pI = 3.6) C-terminus (643-1130aa), in which ~50% of the amino acids are proline or glutamate residues. PELP1's C-terminus is also poorly conserved in eukaryotes and computational predictions support a high degree of disorder and instability within this region. In stark contrast, PELP1's N-terminal region Rix1 domain (1-642aa) is highly conserved and produces high-confidence ordered structural predictions[29] (Supplementary Note 1, Supplementary Fig. 1a). Due to the instability of the C-terminus, we designed a FLAG-tagged PELP1 expression construct coding for the N-terminal Rix1 domain of PELP1 (1-642aa). We then repeated the Rix1 complex reconstitution IP experiment using the PELP1 Rix1 domain as bait to see how the lack of PELP1 C-terminus affects Rix1 complex assembly. Our results indicated that the conserved Rix1 domain of PELP1 stably associates with WDR18 but not TEX10 or SENP3 (Fig. 1b, lanes 8–10).

Since PELP1's C-terminus is required for full Rix1 complex assembly, we performed reconstitution and IP with a C-terminal FLAG-tagged PELP1 to enrich for complexes associated with full-length PELP1. These results were consistent with the N-terminal IP results, but we were able to observe more stoichiometric amounts of TEX10 and SENP3 (Supplementary Fig. 2a). Interestingly, we still observed the PELP1 C-terminal degradation band (albeit in lesser amounts). This is suggestive of a higher-order stoichiometry in which more than one PELP1 protomer exists in the Rix1 complex. In contrast to the N-terminal PELP IP, the C-terminal PELP1 IP appeared to better associate with TEX10 when lacking SENP3, and addition of SENP3 results in a decrease of C-terminal PELP1 degradation (Supplementary Fig. 2a, lanes 4 vs. 5). This supports a revised view that TEX10 can stably associate with the C-terminal region of PELP1 and that SENP3 provides molecular stability to this region of the Rix1 complex. To further probe the interactions within the Rix1 complex we created a series of PELP1 C-terminal truncations and then performed reconstitution and IP to determine which regions to PELP1 are required for TEX10 and SENP3 association (Fig. 1c). Both TEX10 and SENP3 still associate with PELP1 upon removal of residues 802-1130 from PELP1's C-terminus, revealing that the acidic Asp/Glu-rich region of PELP1 is not required for Rix1 complex formation. Interestingly, the C-terminal half of PELP1 (643-1130aa) is sufficient for SENP3 association but not TEX10. The

association of SENP3 with the C-terminal half of PELP1 is supportive of earlier work demonstrating that PELP1[K826] is the primary SUMO conjugation site within PELP1[16]. The N-terminal Rix1 domain is sufficient for WDR18 binding and we do not detect WDR18 binding to the isolated PELP1 C-terminal domain. Taken together our data support PELP1 being the core structural component of the Rix1 complex with aided C-terminal stability by TEX10 and SENP3 (Fig. 1d). The conserved Rix1 domain of PELP1 forms a stable sub-complex with WDR18, while the flexible C-terminus of PELP1 forms a sub-complex with SENP3. In contrast TEX10 relies on both the N- and C-terminal domains of PELP1 for binding.

## Crosslinking mass spectrometry of human Rix1 complex

Given our ability to isolate stoichiometric amounts of the Rix1 complex, we pursued crosslinking mass spectrometry (MS) experiments to map interactions between the four protein members of the complex. The Rix1 complex was purified via the C-terminal FLAG tag on full length PELP1 and then concentrated prior to crosslinking. The small molecule BS3 was used to covalently crosslink primary amines from lysine residues within the complex. We tested a gradient of BS3 to determine the optimal concentration of 0.12 mM BS3 for subsequent MS analysis (Supplementary Fig. 3a and Supplementary Data 1). The crosslinked complex was quenched with Tris buffer and then sub-

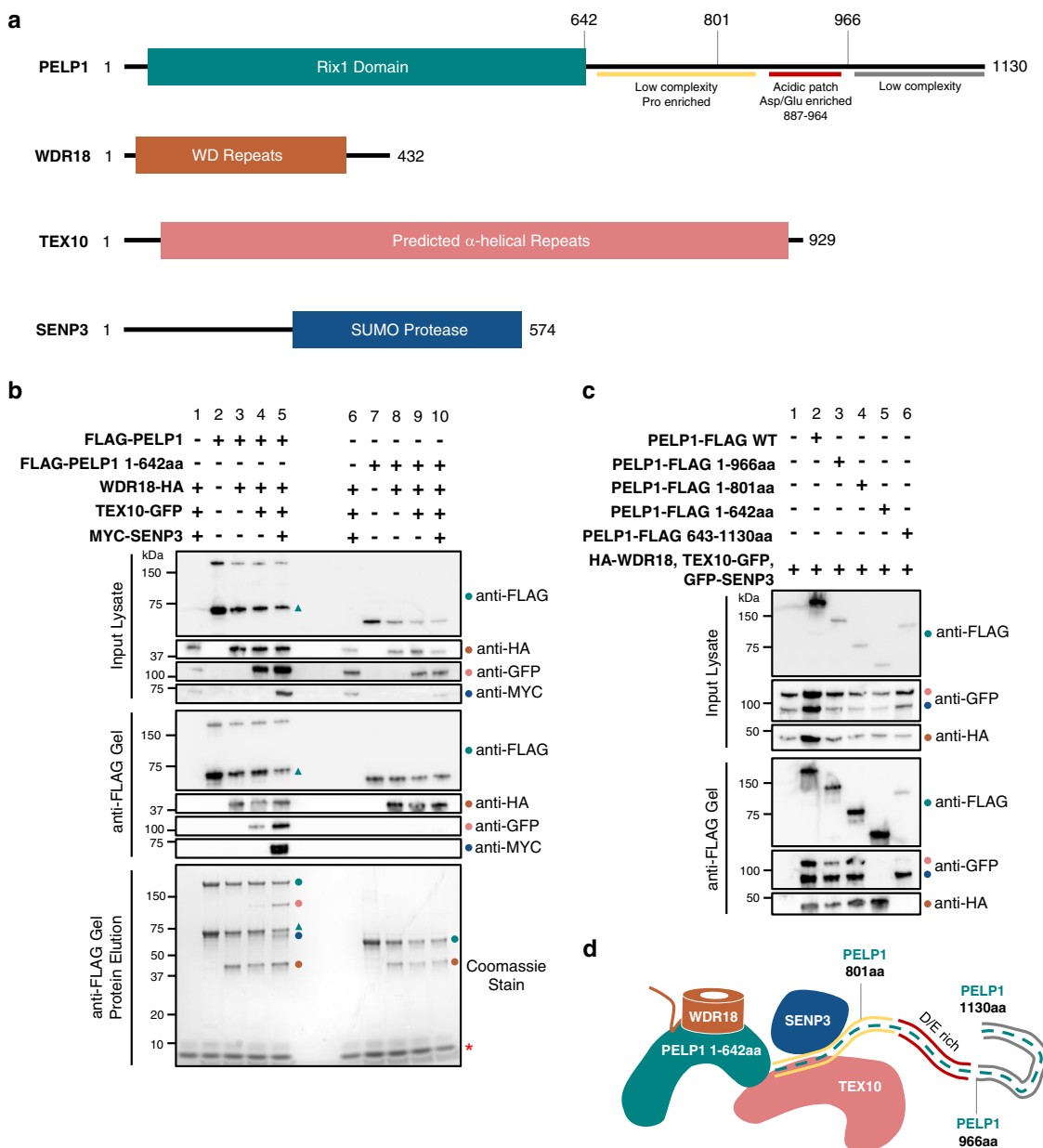

**Fig. 1 | Reconstitution of the human Rix1 complex. a** Schematic representation of human Rix1 complex members PELP1 (teal), WDR18 (orange), TEX10 (pink), and SENP3 (blue) with amino acid boundaries. **b** Western blot and total protein stain from SDS-PAGE of sequentially built reconstituted Rix1 complexes isolated by co-IP of N-terminal FLAG tagged PELP1 constructs. Lanes 1-5 are Rix1 complexes built over full-length PELP1 (1-1130aa) and lanes 6-10 are complexes built over PELP1 Rix1 domain (1-642aa). Teal triangle denotes band representing C-terminal truncated PELP1 originating from full-length. Colored circles denote bands representing like- colored Rix1 complex members in (**a**). Red asterisks denote 3X FLAG peptide band. **c** Western blot from SDS-PAGE of the human Rix1 complexes affinity isolated using PELP1 C-terminal truncations for mapping of TEX10 and SENP3 binding regions within the PELP1 C-terminus. **d** Cartoon representation of human Rix1 complex members interacting with proteins/regions determined in (**b**, **c**). co-IP and Western blot experiments in (**b**, **c**) were performed at least two times with reproducible results. Uncropped gel images available in Source Data file.

jected to tryptic digestion and MS. An analysis of the results revealed intra-molecular crosslinks between the individual subunits (Supplementary Fig. 3b, purple lines) as well as inter-molecular crosslinks between all members of the complex (Supplementary Fig. 3b, green lines). In addition, we performed control experiments where no BS3 was added. While no false-discovery rate was calculated by the search engine, we identified a single spectrum as a false positive from the control samples when using the cutoff parameters described in the methods but found 99 spectra that matched to putative crosslinks in the Rix1 complex when BS3 was used to crosslink the samples. While our IP analysis revealed that the C-terminal half of PELP1 is required for TEX10 and SENP3 binding we could not detect any crosslinks present in the C-terminal region of PELP1. We attribute this result to the unusual amino acid composition of this region (643-1130aa) and the limitations associated with using an amine-reactive crosslinker. The C-terminus of PELP1 only contains 6 lysine residues, most of which reside in trypsin cleavage sites that are predicted to have less than 100% cleavage probability (Expasy, PeptideCutter tool), thus reducing ability for detection. We detected several crosslinks between the Rix1 domain of PELP1 and WDR18, which is in good agreement with our IP results. We detected a few crosslinks from the N-terminus of TEX10 with the PELP1 Rix1 domain, supporting our IP results that showed the Rix1 domain is required for binding. We also detected numerous intra-molecular crosslinks within the predicted structured HEAT-repeat region of TEX10. Finally we observed many crosslinks between the predicted disordered N-terminal domain of SENP3 with PELP1, TEX10, and WDR18, suggesting that this domain is very flexible in solution (Supplementary Fig. 3b and Supplementary Data 1).

Next we performed BS3 crosslinking MS analysis on the stable PELP1 Rix1 domain and WDR18 sub-complex. The complex was purified via the N-terminal FLAG-tagged PELP1 Rix1 domain. We used a gradient of BS3 to determine the optimal concentration for MS analysis (Supplementary Fig 4a). Due to the enhanced stability of the subcomplex we were able to use a higher protein concentration for MS, resulting in the detection of more crosslinks than with the full complex. Again, we performed control experiments where BS3 was not added. In these samples, when using the cutoff parameters described in the methods, we did not observe any false positives but identified 94 spectra that matched to putative crosslinks in the PELP1 Rix1 domain and WDR18 sub-complex when BS3 crosslinked. Like the full complex, we observed many of the same intra-molecular PELP1 crosslinks suggesting that the binding of SENP3 and TEX10 does not alter the structure of the PELP1 Rix1 domain significantly. We observed many of the same inter-molecular crosslinks between PELP1 and WDR18, also suggesting that SENP3 and TEX10 binding does not alter the interfaces between PELP1 and WDR18 (Supplementary Fig. 4b, c). Collectively the BS3 crosslinking supports that the interfaces formed between PELP1 and WDR18 in the stable subcomplex are retained in the full Rix1 complex.

## Cryo-EM structure of human PELP1-WDR18

We determined the cryo-EM structure of the stable subcomplex comprised of the PELP1 Rix1 domain and full-length WDR18. The PELP1-WDR18 complex was purified from large-scale transient transfection in HEK293FT cells. Cryo-EM analysis of the isolated PELP1-WDR18 subcomplex revealed particles roughly 100 Å in diameter with apparent C2 symmetry as seen by 2D image classification (Fig. 2b). Single-particle reconstruction resulted in a 2.7 Å resolution map composed of a C2 symmetric PELP1-WDR18 heterotetramer assembly (Fig. 2c, Supplementary Figs. 5, 6). Alphafold predicted models for the PELP1 Rix1 domain and WDR18 were docked into the reconstruction and used as initial models for model building and refinement (Fig. 2d, Supplementary Fig. 1a, b). Overall, the Alphafold predicted models are in good agreement with the final refined structure, however, we did observe a major re-arrangement of the four N-terminal helices from the PELP1 Rix1 domain. The refined structure reveals a highly interconnected

tetrameric assembly centered around two dimerized PELP1 Rix1 domains, which form an α-solenoid structure resembling a horseshoe. Dimerization of the two Rix1 domains creates a hollow core within the center of the tetramer. The two β-propeller domains of WDR18 sit atop and sideward to the dimerized horseshoes. The C-terminus of WDR18 threads through the center of the hollow core towards the bottom of the assembly (Fig. 2c, d). The arrangement of the PELP1 and WDR18 subunits within the structure is also supported by the BS3 crosslinking analysis. The N-terminus of PELP1 is in close proximity with the C-terminus of WDR18, while the C-terminal half of the Rix1 domain is near WDR18's N-terminal region. Finally, the overall human PELP1-WDR18 subcomplex bares remarkable structural similarities to the orthologous yeast subcomplex (RIX1-IPI3) previously identified bound to the pre-60S ribosome (Supplementary Fig. 7a–d)[30]. One notable difference is that the ribosome-bound RIX1-IPI3 tetramer is asymmetric suggesting that the binding of IPI1 (TEX10) and/or the ribosome induces asymmetry.

While the overall structural assembly of this subcomplex appears highly conserved in eukaryotes, the biologically relevant PxxP and LxxLL sequence motifs present in human PELP1-WDR18 are not (Supplementary Note 1). The LxxLL motifs in PELP1 do not begin to appear in evolution until vertebrates, consistent with phylogenetic work on steroid receptor signaling evolution[31,32]. Our cryo-EM structure allowed visualization of all eleven LxxLL motifs and one of three PxxP motifs embedded in PELP1 (Fig. 2d–f). The two PxxP motifs we were not able to visualize reside in disordered regions of the Rix1 domain that extend out as breaks in the α-solenoid structure. The one modeled PxxP (PM2) is structured within an α-helix and adjacent to the LxxLL motif LM2 (Fig. 2e, f). Ordered LxxLL motifs have a high propensity to reside in α-helical secondary structures, which we observed for all eleven LxxLL motifs in PELP1's Rix1 domain (Fig. 2d, f). Seven of PELP1's LxxLL motifs are localized on the internal (nonsolvent) face of the PELP1-WDR18 assembly, while the remaining four motifs are solvent exposed (Fig. 2f). The structural localization of these binding motifs has significant implications on their functionality, mainly their ability to interact with protein binding partners.

## PELP1 LxxLL motifs facilitate PELP1-WDR18 inter-subunit bridging

The majority of the LxxLL motifs within the structure facilitate inter-subunit bridging within the complex. The most studied functional role of the LxxLL motif is to mediate protein-protein interactions between transcriptional regulators through non-polar van der Waals forces[33,34]. Previous structures of proteins that bind LxxLL motifs exhibit an exclusive hydrophobic patch recognized by the LxxLL motif and flanking residues (Supplementary Fig. 8)[34–37]. The structure of the PELP1-WDR18 subcomplex revealed that the majority of PELP1's LxxLL motifs play direct roles in mediating subcomplex formation through hydrophobic inter-subunit interactions between LxxLL motifs and ordered helices. This interaction mode is highly exhibited between the PELP1 Rix1 domain dimer that acts as the core of the PELP1-WDR18 assembly. This dimerization is mediated through three interfaces, all of which contain LxxLL motifs and are positionally influenced by the C2 symmetry of the overall assembly (Fig. 3b, d). The first two interfaces, which are identical due to symmetry, form a tri-LxxLL interaction between the PELP1 protomers (LM8[1]-LM1[2]-LM2[2], superscripts denoting individual PELP1 protomers) and is dictated by non-polar leucine residues (Fig. 3b, c). The positioning of these motifs within each PELP1 protomer has the dimer interfaces formed between the middle α14 helix in the inward-most bend of the first horseshoe and the most N-terminal helical repeat (α1 + α2) of the second protomer horseshoe (Fig. 3b). The LM8[1] LxxLL motif of PELP1 resides in the beginning of α14, a region filled with non-polar leucine residues. This non-polar stretch of residues containing the LM8 motif lies across the surface of α1 and α2 of the neighboring protomer, which contain the

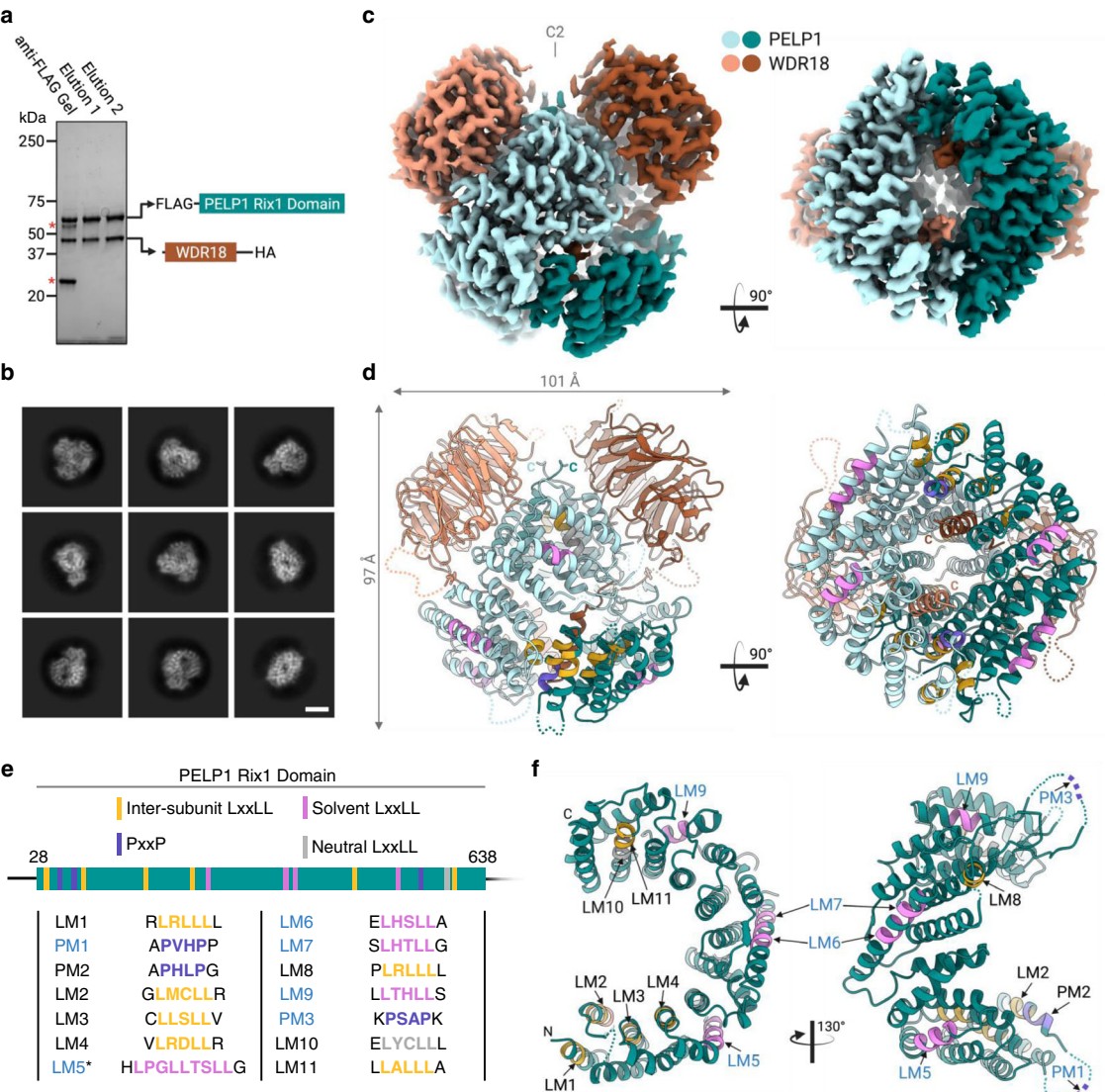

**Fig. 2 | Cryo-EM structure of the human PELP1 Rix1 domain bound to WDR18. a** SDS-PAGE and total protein stain of purified PELP1-WDR18 sub-complex from HEK293FT cells. Red asterisks denote IgG bands from anti-FLAG affinity gel. Protein purification was performed at least three times with reproducible results. **b** Representative single-particle 2D classification images from cryo-EM data processing. Scale bar = 50 Å. **c** Sharpened cryo-EM density of PELP1 Rix1 domain bound to WDR18 exhibiting a heterotetramer assembly with C2 symmetry. Side and bottom views are shown. **d** Ribbon model of same views shown in (**c**). The visualized binding motifs (LxxLL and PxxP) in PELP1's Rix1 domain are color-coded based on solvent accessibility and roles within the assembly. Magenta = solvent LxxLL, Gold = inter-subunit bridging LxxLL, Gray = neutral (neither solvent nor bridging) LxxLL, Purple = PxxP. **e** Schematic representation of PELP1's Rix1 domain with localization of LxxLL and PxxP motifs. Motifs are color-coded as in the model in (**d**). Each motif's amino acid sequence along with flanking residues. Asterisk on LM5 denotes an overlapping LxxLL sequence register. **f** Individual PELP1 Rix1 domain protomer view of LxxLL and PxxP motifs localized within the structure. Blue-lettered motif labels denote solvent-accessible motifs. Model coloration is consistent with (**d**,**e**).

leucine-contributing LM1[2] and LM2[2] LxxLL motifs, respectively. These non-polar Rix1 domain dimerization interfaces appear to be dynamic as evidenced by their lower local resolution. To further investigate the dynamics we performed 3D variability analysis of the cryo-EM map and observed that this region of PELP1 exhibits conformational heterogeneity (Supplementary Fig. 5f, Supplementary Movie 1). Helices α1 and α2 of PELP1 appear to be the most flexible/dynamic. During the variability analysis, we observed that the density for these helices disappears into disorder (Supplementary Movie 1).

The overall architecture of these contact points between PELP1 protomers appears to be conserved from yeast, especially the contributions from the N-terminal α1 and α2 helices (Supplementary Fig. 9a, c). However, the contributing sequence and structural factors from the inward bend of the Rix1 domain horseshoe differ slightly. Instead of α14 contributing solely from one PELP1 protomer to

facilitate dimerization, yeast appears to require an additional helix/ loop (α12) to promote the interface (Supplementary Fig. 9c). The enriched non-polar chemical nature of α14/LM8 in mammalian PELP1 is not conserved in yeast, presumably leading to this need for more structural contribution to promote Rix1 domain dimerization.

The third PELP1 dimer interface lies along the axis of symmetry and is dictated by non-polar residue interactions (Fig. 3d, e). This interface relies on a grouping of four α-helices (α20 and α22 from each PELP1 protomer) that reside at the most C-terminal end of the Rix1 domain horseshoe fold. Both α20 and α22 helices contain some of the most conserved non-polar residues in PELP1's Rix1 domain and are likely crucial for the conservation of the dimerized assembly (Supplementary Fig. 9c). Two tandem LxxLL motifs (LM10 and LM11) make up more than half of the residues in the α20 helix. LM11, which is the final LxxLL motif within PELP1, is positioned at the top of α20 where its

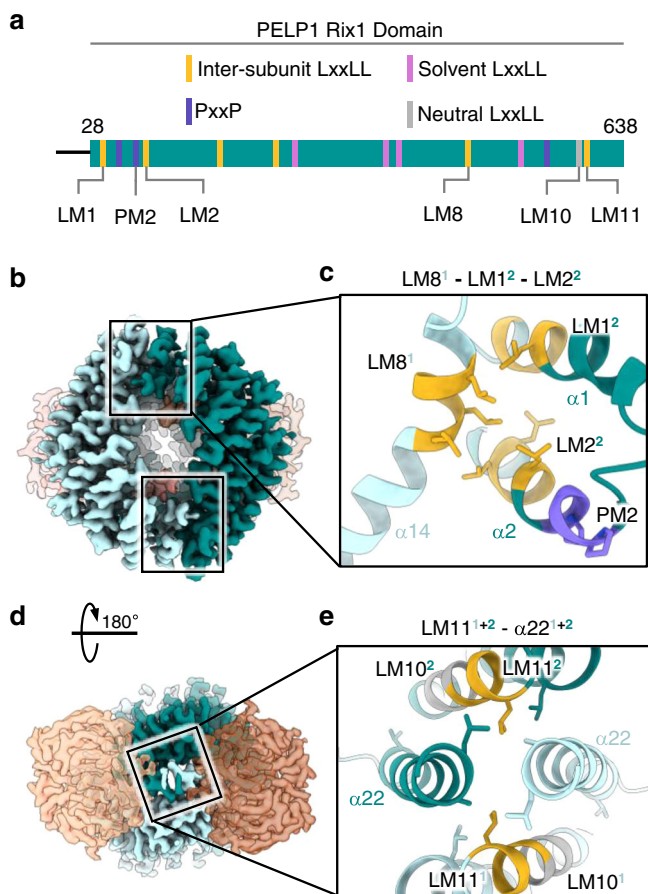

**Fig. 3 | Structural basis of PELP1-PELP1 dimerization. a** Schematic representation of PELP1's Rix1 domain with localization of LxxLL and PxxP motifs. Specific motifs involved in PELP1 dimerization, or residing close to dimer interfaces, are noted below the schematic. **b** Bottom view of segmented cryo-EM density showing two symmetric PELP1 dimerization interfaces between LM8¹-LM1²-LM2² (superscript denotes specific protomer). **c** Model zoom of LM8¹-LM1²-LM2² dimer interface exhibiting a hydrophobic environment produced by mainly leucine residues from LxxLL motifs. **d** Top view of cryo-EM density showing a single symmetric PELP1 dimerization interface between LM11 and α-helix 22 of each PELP1 protomer. **e** Model zoom of LM11¹⁺²-α22¹⁺² interface showing contributing leucine residues to another hydrophobic interface environment.

highly conserved L588 contributes to the non-polar dimer interface (Fig. 3e). LM10 is the only LxxLL motif in PELP1 that is non-solvent accessible but does not play a direct role in PELP1 dimerization. For this reason, we have deemed LM10 as a "neutral" LxxLL motif within PELP1. However, its role in providing helical structure to α20 likely provides indirect integrity to the dimer interface.

The PELP1 dimer formed by the Rix1 domains creates the scaffold upon which two WDR18 protomers bind. Unlike PELP1, we do not observe the two WDR18 protomers contacting each other in the structure. Each WDR18 protomer makes extensive contacts with both copies of PELP1 in the assembly. These interactions are best observed at the interface between the bottom of the seven-bladed β-propeller domain of WDR18 and the top dimerization interface of PELP1's Rix1 domains (Fig. 4c–e). Six of the seven propeller blades contribute loops that interact with PELP1 and appear to be split amongst the two protomers of PELP1 that are dimerized within the interface (Fig. 4e). Many of the WDR18 residues provided from these loops are non-polar and appear to contribute to the highly hydrophobic area of the C-terminal dimerization interface between PELP1 at the top of the assembly (Fig. 4d). This suggests that WDR18's propeller domain acts as a molecular "cap" that further

promotes and stabilizes dimerization between PELP1's Rix1 domains.

One of the more noticeable features of the PELP1-WDR18 assembly is the C-terminal tail of WDR18 that enters through an external gap in the side of the PELP1 scaffold and stretches through the hollow center of the core. This tail contains an ordered α-helix (WDR18 α3) that participates in further interactions with both PELP1 protomers (Fig. 4f, g). Most of these interactions take place between WDR18 α3 and three PELP1 α-helices within the same protomer that contain LxxLL motifs LM2, LM3, and LM4. These three motifs encompass α3 of WDR18 and provide important non-polar and polar interactions for α3 placement (Fig. 4g). WD-repeat proteins are normally highly conserved in their β-propeller domains and may or may not have unique extensions on either terminus. Outside of WDR18's β-propeller domain, the ordered density for the C-terminus that travels through the PELP1 scaffold core also appears to be conserved, especially within the α3 helix (Supplementary Note 2). This points to the C-terminal tail likely being important for PELP1-WDR18 assembly. Further, the 3D variability analysis of the cryo-EM map exhibits dynamics within the WDR18 C-terminal tail that mimics the N-terminal helices of PELP1 (Supplementary Movie 1). The orthologous yeast assembly exhibits an identical placement of the WDR18/IPI3 C-terminal tail, signifying that this is indeed a conserved structural feature (Supplementary Fig. 9b, d). To confirm the significance of this interaction we constructed a WDR18 mutant in which the C-terminal tail was deleted, leading to the expression of a protein comprised only of WDR18's seven-bladed β-propeller (1-344aa). Using the WDR18 mutant as bait to assess its ability to bind PELP1, we found that deletion of the C-terminal tail drastically reduced the ability for PELP1 association (Fig. 4h). Weak signal for only the truncated PELP1 species was detectable upon IP of C-terminal deleted WDR18. These observations suggest that the β-propeller domain of WDR18 is not sufficient for binding to PELP1. Instead, this supports a mechanism in which WDR18's C-terminal tail plays a required structural role in anchoring WDR18 to PELP1 and then fostering PELP1 dimerization through "capping" by the β-propeller domain. These results are consistent with experiments carried out with the yeast Rix1 complex, which revealed that the C-terminal helix of Ipi3 (WDR18) is required for the oligomerization of the complex[38].

## PELP1's solvent LxxLL motifs are not positioned for steroid receptor binding

An unexpected observation from the structure was that none of PELP1's LxxLL motifs are in a conformation that would support steroid receptor binding. We were able to visualize all eleven LxxLL motifs present in PELP1, along with one of three PxxP motifs. The single PxxP motif we localized (PM2) resides in helix α2 of PELP1 on the non-solvent face of the assembly. Considering the inaccessibility of this motif to binding factors, we reason that this motif is incompatible with SH2 domain binding and is likely only important for structural integrity for the α2 helix that comprises one of the PELP1 dimerization interfaces. We were not able to visualize the remaining two PxxP motifs due to their localization in flexible loops within the structure. However, because these loops protrude toward the solvent face, it is likely that these motifs could support SH2 domain binding.

Only four of PELP1's LxxLL motifs are solvent exposed, while the other seven facilitate higher-order assembly of PELP1-WDR18. Along with internal and flanking sequence composition, the solvent accessibility of an LxxLL motif is one of the most important determining factors for its ability to bind and co-regulate steroid receptors[34]. The binding mechanism between an LxxLL motif of a coregulator and a steroid receptor takes place through largely non-polar interactions within the activation function-2 (AF-2) region of the steroid receptor's ligand binding domain. This AF-2 region has a hydrophobic pocket common amongst all nuclear steroid receptors and directly binds the LxxLL sequence, heavily utilizing leucine residues in positions 1 and 5

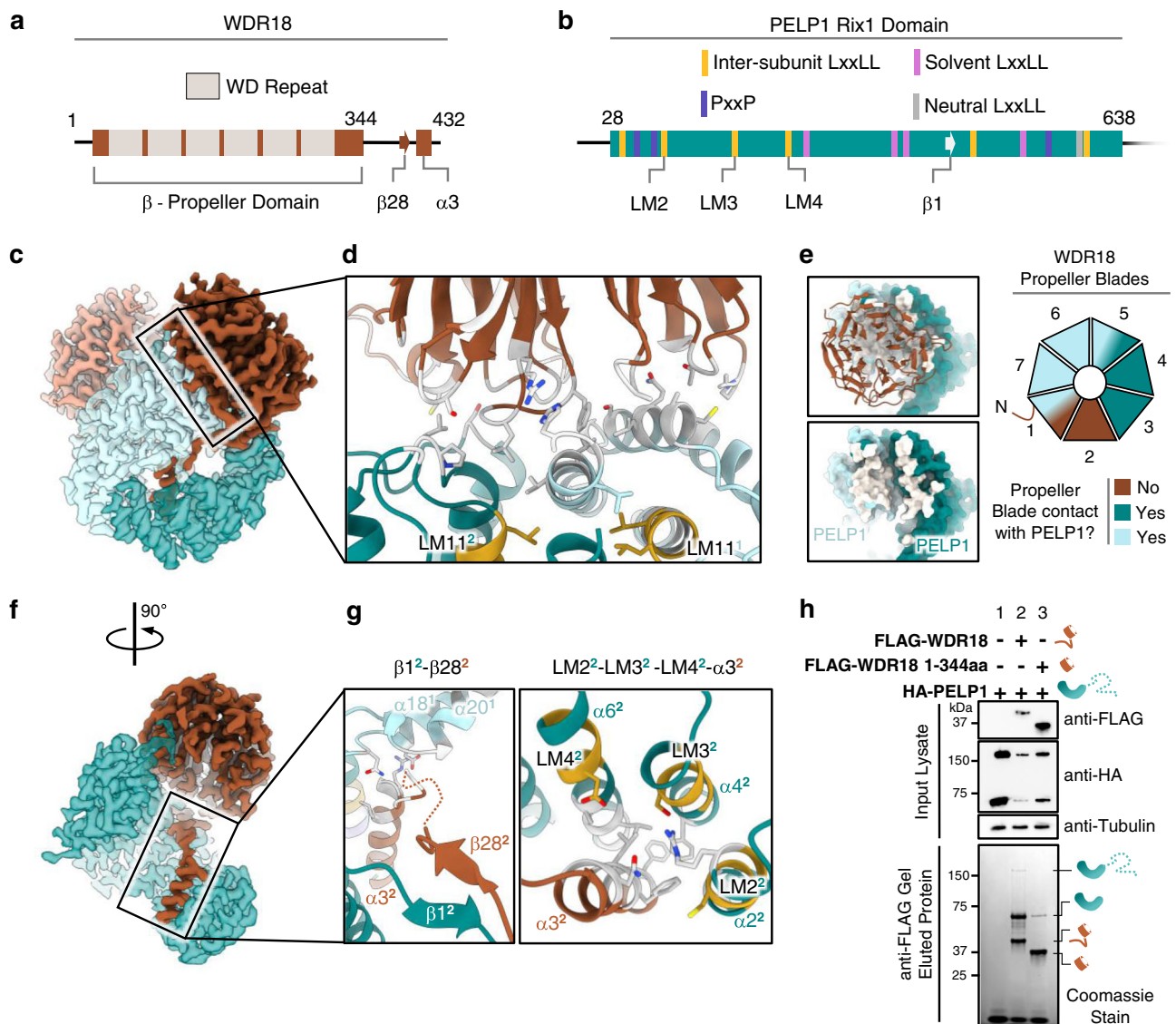

**Fig. 4 | Structural basis of PELP1 Rix1 domain interaction with WDR18.**
**a** Schematic representation of WDR18 with WD repeat regions that contribute to β-propeller domain and features of the C-terminal tail noted below. **b** Schematic representation of PELP1's Rix1 domain with localization of LxxLL and PxxP motifs. Specific motifs and features involved in PELP1-WDR18 interaction are noted below. **c** Side view of cryo-EM density showing interaction plane between dimerized PELP1 and WDR18. **d** Model zoom of WDR18 β-propeller domain loops extending into both protomers of PELP1. Viewpoint is from top of PELP dimer interface (LM11 in view). Parts of the model that are colored white represent residues involved in the PELP1-WDR18 interaction interface. Superscripts denote individual protomers. **e** Footprint representation of WDR18 interaction with both PELP1 protomers in the assembly, illustrated by white colored surface on PELP1. Cartoon representation of the 7 WDR18 β-propeller blades are color coordinated (light blue and teal) to exhibit individual blade contacts with specific PELP1 protomers. Brown coloration stands for no blade contact with PELP1. **f** Side view of cryo-EM density with half the

assembly hidden for visualization of WDR18 C-terminal tail inside the hollow core. **g** Model zooms of WDR18 C-terminal tail interactions with inner surface of PELP1. Left panel illustrates small β-sheet interactions between β1 of PELP1 Rix1 domain and β28 of the WDR18 C-terminal tail as it descends into the hollow assembly core. Right panel illustrates a descending α-helix 3 of the WDR18 C-terminal tail interacting with three LxxLL motifs of the same PELP1 protomer (LM2-LM3-LM4). Parts of the model that are colored white represent residues involved in the PELP1-WDR18 interaction interface. LxxLL motifs colored in gold to represent roles in assembly inter-subunit bridging. Superscripts denote individual protomers. **h** N-terminal FLAG-tagged WDR18 wild type or C-terminal tail deleted mutant (1-344aa) constructs were used as bait for co-IP and assessed for full-length PELP1 binding. SDS-PAGE of eluted co-IP samples followed by total protein staining exhibited enrichment of C-terminally truncated PELP1 only with wild type WDR18 and not with C-terminal tail deleted WDR18 (lane 2 vs. 3). co-IP and Western blot experiments in (**h**) were performed at least two times with reproducible results.

---

of the motif (Supplementary Fig. 8)[34–37]. The four LxxLL motifs that reside on the solvent face of the assembly include LM5, LM6, LM7, and LM9 (Fig. 5a, b). LM4 and LM5 have previously been suggested to support ERα coactivation[4]. Interestingly, each of the four solvent-exposed LxxLL motifs in PELP1 had their leucine residues buried away from the solvent surface, most of which reach back into the PELP1 structure to facilitate α-helical arrangements (Fig. 5c–e). Therefore, we reason that these PELP1 LxxLL motifs are not compatible with steroid receptor binding via the AF-2 mechanism, raising

questions about PELP1's ability to coregulate steroid receptors in the context of the PELP1-WDR18 assembly.

## WDR18 association with PELP1 suppress PELP1-mediated ERα transcription in reporter assays

Because the structure suggests none of PELP1's LxxLL motifs appear suitable for receptor interaction, we sought to investigate if the PELP1-WDR18 assembly associates with ERα. We pursued endogenous co-IP experiments from MCF7 cells in which PELP1, WDR18, and ERα are well

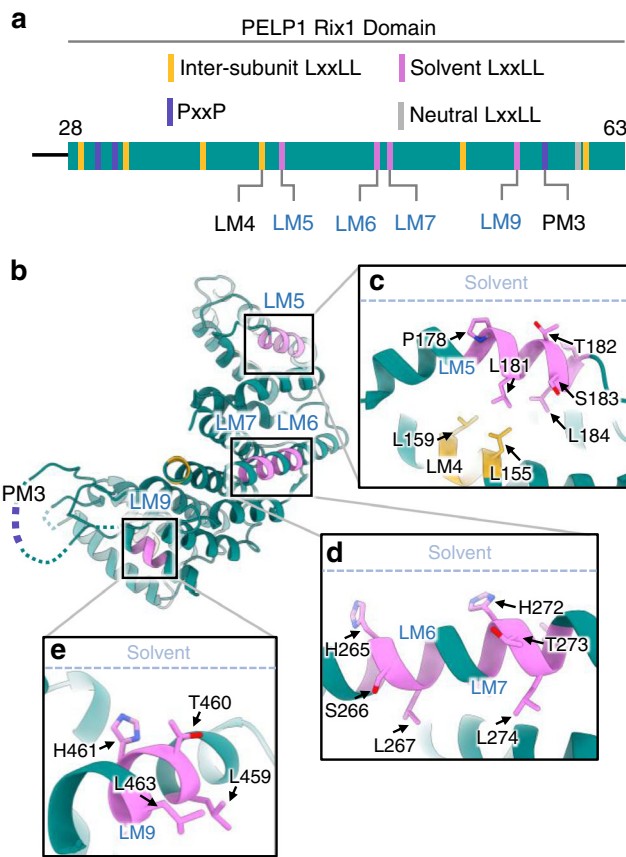

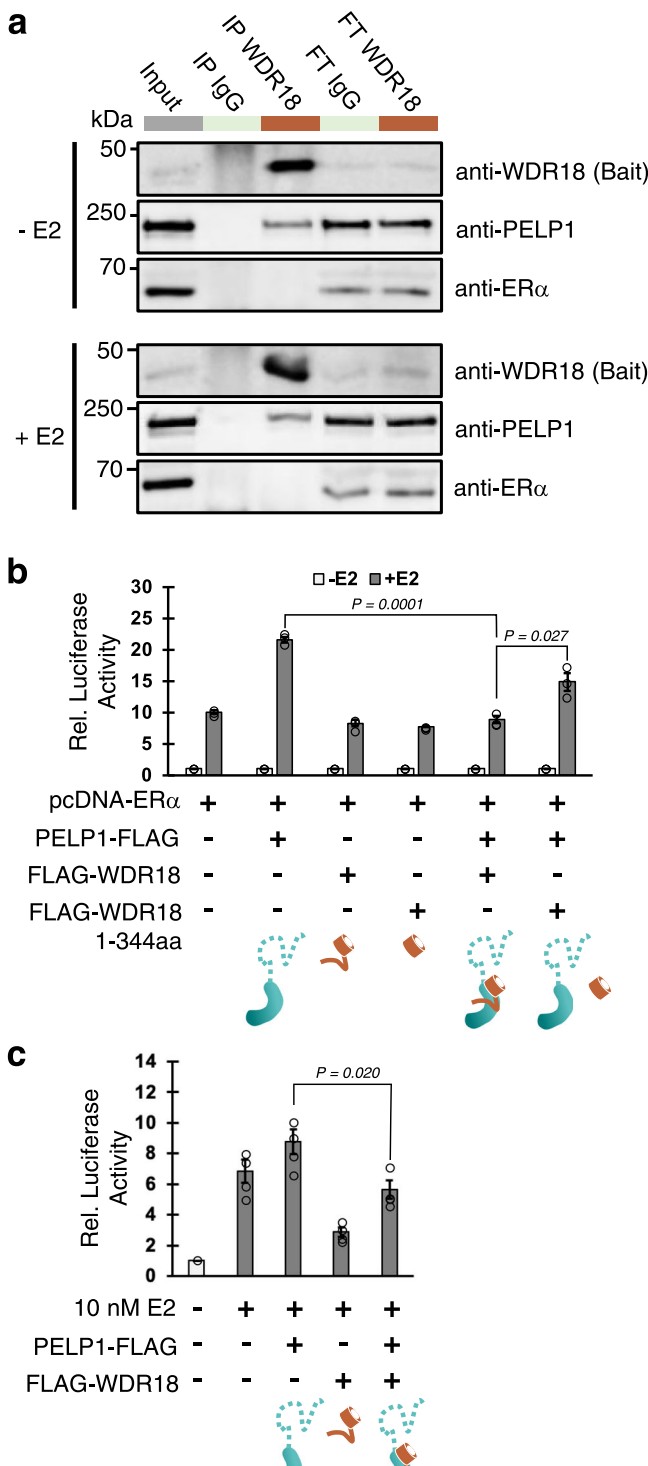

**Fig. 5 | PELP1's solvent-exposed LxxLL motifs are incompatible with steroid receptor binding. a** Schematic representation of PELP1's Rix1 domain with localization of LxxLL and PxxP motifs. Motifs illustrated in model views are noted below the schematic (solvent motif labels colored blue). **b** Model view of a single PELP1 Rix1 domain protomer with solvent-exposed LxxLL motifs colored magenta. **c**, **d**, **e** Individual zooms of each solvent LxxLL motif with residue positions displayed. Solvent face is illustrated in each panel with dashed line for spatial orientation of the motifs. The invariant leucine residues required for AF-2 binding of SRs are buried away from the solvent face, rendering them inaccessible for SR binding.

expressed. Co-IP of endogenous WDR18 exhibited a clear enrichment for PELP1 upon Western blotting, indicating the isolation of endogenous PELP1-WDR18 complexes (Fig. 6a). However, we did not observe signal for ERα in the WDR18 co-IP lane, even after cell treatment with estrogen (Fig. 6a). This supports our structure-informed hypothesis that the PELP1-WDR18 assembly does not support the association with ERα. We also did a reciprocal co-IP of endogenous PELP1 to assess the enrichment of WDR18 and ERα. Western blotting resulted in signal for abundant WDR18 enrichment upon PELP1 IP, and only slightly detectable enrichment of ERα in plus and minus estrogen conditions (Supplementary Fig. 10). This differing enrichment ratio upon endogenous PELP1 IP is suggestive of the PELP1-WDR18 assembly being the predominant formation in the cell over ERα interaction. A recent sensitive quantitative proteomics approach did not identify PELP1 as one of the most frequently enriched ERα interactors in MCF7 cells, suggesting that PELP1 is not a primary interaction partner of ERα in MCF7 cells[39]. Moreover, the association between ERα and PELP1 may be very transient, cell type specific, or sensitive to the IP conditions used making it difficult to capture. We also cannot exclude the possibility that the interaction between ERα and PELP1 is indirect, given the large number of PELP1 interaction partners that have been identified[3].

Finally, we investigated if the components of the PELP1-WDR18 assembly can facilitate a hormone-dependent ERα transcriptional response. We utilized the well-established dual luciferase reporter

assay to measure the effects of PELP1-WDR18 assembly members on 3X estrogen response element (ERE) promoter activity transfected in HepG2 cells. Transient overexpression of PELP1 alone was sufficient to promote a significant estrogen-dependent ERα transcriptional response, consistent with previous reports[4,8] (Fig. 6b, Supplementary Fig. 11a). WDR18 alone failed to promote a response, indicating WDR18 does not play a coactivating role (Fig. 6b). However, co-expression of wild type PELP1 and WDR18 resulted in no elevated transcriptional activation compared to control, suggesting that WDR18 association with PELP1 prevents PELP1-meidated coactivation of ERα (Fig. 6b). We followed up this hypothesis with one more co-expression condition

**Fig. 6 | WDR18 association with PELP1 reduces ERα transcription coactivation.**
**a.** Endogenous co-IP of WDR18 from MCF7 cells treated with 10 nM E2 or ethanol. Western blot shows WDR18 interaction with PELP1 but not ERα. FT, flow through. co-IP and Western blot experiment in (**a**) was performed three times with reproducible results. **b** Dual luciferase assays in HepG2 cells treated with 10 nM E2 or ethanol showing ERα mediated transcription at a 3xERE promoter (*n* = 3 biologically independent samples per experiment). Overexpression of PELP1 induces robust transcriptional coactivation of ERα while WDR18 wild type and C-terminal deleted mutant (1-344aa) does not. Co-expression of PELP1 with WDR18 wild type prevents coactivation of ERα. Co-expression of PELP1 with WDR18 1-344aa regains ability for ERα coactivation due to impaired binding ability between PELP1 and WDR18 1-344aa. Independent experiments with *n* = 3 was performed three times with representative results shown. Statistics were determined using n from the one shown result. **c** Dual luciferase assays in MCF7 cells relying on endogenous ERα machinery with a similar experimental setup as in (**b**) (*n* = 4 biologically independent samples per experiment). Overexpression of PELP1 induces a detectable transcriptional coactivation of ERα while WDR18 wild-type overexpression alone reduces coactivation below empty vector control. Co-overexpression of PELP1 and WDR18 significantly reduces ERα coactivation below that of PELP1 overexpression alone. Independent experiments with *n* = 4 was performed two times with representative results shown. Statistics were determined using n from the one shown result. Source data are provided in the Source Data file.

using WDR18 that lacks its C-terminal tail which does not bind robustly to PELP1 (Fig. 4h). If WDR18 interaction with PELP1 is preventing coactivation, then reduced WDR18 association should promote PELP1 coactivator functions. Upon co-expression of wild-type PELP1 and C-terminal tail deleted WDR18 we observed a significant increase in transcription activity over both control and wild-type co-expressed PELP1-WDR18 (Fig. 6b). We repeated the dual luciferase assay in MCF7 cells where we relied on endogenous ERα to stimulate a transcriptional response. While the estrogen-dependent signal was lower in the MCF7 cells the overall trends observed with the HepG2 cells remained the same. PELP1 overexpression alone led to a detectable transcriptional response that was reduced upon co-expression of PELP1 and WDR18 (Fig. 6c, Supplementary Fig. 11b). Together these observations, taken in context with earlier presented data, supports a role for WDR18 in preventing PELP1 coactivation of ERα, likely by holding the LxxLL motifs of PELP1 in a conformation that does not favor SR coactivation. However, additional experiments, will be needed to fully establish if WDR18 binding to PELP1 blocks ERα coactivation in vivo.

## Discussion

PELP1 serves as the central scaffold for the Rix1 complex (PELP1-WDR18-TEX10-SENP3) which plays fundamental roles in ribosome synthesis and heterochromatin maintenance[14,15,18,21,30,40]. In this study, we reconstituted the human Rix1 complex and discovered that WDR18 forms a stable sub-complex with the Rix1 domain of PELP1. We determined a 2.7 Å resolution structure of PELP1's Rix1 domain bound to WDR18, which revealed a highly interconnected heterotetramer assembly which is structurally similar to the *S. cerevisiae* pre-60S bound RIX1-IPI3 tetramer. The structure led to the observation that PELP1's nuclear receptor box motifs (LxxLL), which are not present in lower eukaryotes, are not positioned for SR binding within the PELP1-WDR18 assembly. Finally, we assessed PELP1 coactivation of ERα in the presence of WDR18 through a well-established reporter assay and determined that WDR18 decreases the SR coactivation abilities of PELP1. This work supports a regulatory model of PELP1-mediated SR coactivation where higher-order assembly within the Rix1 complex removes PELP1 from the pool of SR coactivators in the cell (Fig. 7).

Our Rix1 complex reconstitution, crosslinking MS, and cryo-EM structure, reinforce the notion that PELP1 is the central scaffold for the Rix1 complex. WDR18 also plays important scaffolding roles within the Rix1 complex, by providing structural support for PELP1 dimerization. While structural information for the C-terminal region of PELP1 is not

known, this region begins along the top of the PELP1 dimer interface (LM11$^{1+2}$ - α22$^{1+2}$) between the two copies of WDR18, and likely protrudes outward to make further contacts with TEX10 and SENP3. We detected crosslinks between WDR18, TEX10, and SENP3, suggesting WDR18 is close to these complex members. The stoichiometry of TEX10 and SENP3 within the human Rix1 complex has not yet been confirmed. Structural and biochemical data from studies of the *S. cerevisiae* Rix1 complex bound to the pre-60S ribosome show a 2:2:1 ratio of the conserved PELP1-WDR18-TEX10 components (Supplementary Fig. 7c, d)[30]. This stoichiometry is likely a conserved feature of the Rix1 complex given the high degree of structural similarity of PELP1-WDR18 between yeast and human (Supplementary Fig. 7a, b). Our IP analysis revealed that TEX10 requires both the N- and C-terminal domains of PELP1 for binding, suggesting that TEX10 requires PELP1-WDR18 dimerization in addition to the PELP1 C-terminus for its association. This supports a putative semi-conserved binding mode between PELP1-WDR18 and TEX10, similar to what is observed in yeast (Supplementary Fig. 7c–e). Notably, human TEX10 is significantly larger than the yeast homolog IPI1, which might explain why PELP1-WDR18 dimerization alone is not sufficient for TEX10 binding, and why additional regions of the PELP1 C-terminus are needed to facilitate Rix1 complex formation in humans compared to yeast (Supplementary Fig. 7e–g). We suspect that SENP3 association within the Rix1 complex has followed the evolutionary acquisition of PELP1 SUMO post-translational regulation and the expansion of PELP1's unique C-terminal region, which contains a SUMO modification site (lysine 826)[15,16]. Indeed, our results exhibit SENP3 associating with the isolated PELP1 C-terminal domain, and within a region of PELP1 (643-801aa) that flanks the lysine 826 SUMO modification site. We propose that the heterotetrameric assembly of PELP1-WDR18 comprises a stable structural core of the Rix1 complex, with PELP1's C-terminal region being required to foster the interaction between TEX10 and SENP3 with the rest of complex (Fig. 7).

Collectively, our work suggests that the assembly of PELP1 with WDR18 (and likely TEX10, SENP3) does not support a SR coactivating function. It has been proposed that PELP1 can ligand dependently interact with ERα through the AF-2 domain[7,8]. Established work on coactivator binding to the AF-2 domain of SRs exhibit a dynamic LxxLL motif binding model, whereby the LxxLL motif is unstructured until it associates with the hydrophobic pocket of the AF-2 domain[33,34,37,41]. This binding mechanism agrees with our model of WDR18 capping stabilizing PELP1's LxxLL motifs in a structured conformation that prevents AF-2 binding. This suggests that PELP1 assembly within the Rix1 complex directs its scaffolding function away from SR coactivation, and instead promotes the Rix1 complex's roles in ribosome synthesis and/or RNA degradation as part of the Rixosome. This raises questions about when the LxxLL motif(s) of PELP1 could facilitate coactivation. Interestingly, we observed the N-terminal PELP1 LxxLL motifs LM1 and LM2 sampling ordered and disordered states upon 3D variability analysis of our cryo-EM data. These dynamics imply that the Rix1 domain of PELP1 has the propensity to change conformations and likely behaves differently when not bound to WDR18. Additionally, it may be possible for PELP1 to facilitate coactivation of SRs indirectly (independent of AF-2 binding), likely through a secondary or tertiary scaffolding mechanism that promotes SR transcription complexes (SRC-p300/CBP)[8,24,25]. Future in vivo-based studies will be needed to fully establish if WDR18 binding to PELP1 is sufficient to prevent SR coactivation.

Localization and/or post-translational modifications of PELP1 in both normal and cancer tissue could play roles in mediating SR coactivation[14–17,42,43]. Previous work using protein-specific antibodies established that PELP1 and the other associated members of the Rixosome are primarily localized to the nucleus and nucleolus, which is consistent with their well-defined roles in ribosome assembly[14]. Moreover, Rixosome localization also appears to be influenced by the

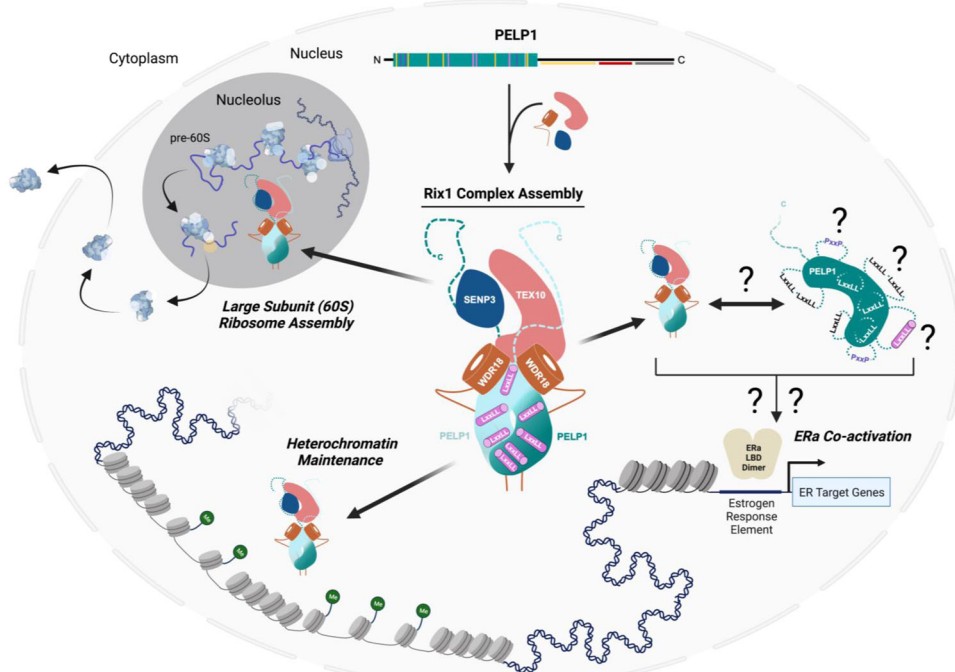

**Fig. 7 | The various biological functions of the human Rix1 scaffolding complex.** The human Rix1 scaffolding complex is implicated in various biological functions in the cell nucleus including large subunit (60 S) ribosome assembly, heterochromatin maintenance, and steroid receptor coactivation of transcription. PELP1 is the central scaffold of the Rix1 complex and is likely primarily associated with, and functions within, this complex in the nucleus during its roles in ribosome assembly and heterochromatin maintenance. PELP1 facilitation/influence on ERα coactivation is likely a secondary or tertiary effect in the heterogenous and functionally diverse ERα signaling pathway. We hypothesize that PELP1 can influence ERα coactivation in transient/unique states, possibly when PELP1 is dissociated from the Rix1 complex either normally or during altered states of PELP1 proteostasis in the cell. This dissociation from the Rix1 complex may allow PELP1 to undergo conformational structural changes that form favorable LxxLL/PxxP motifs allowing for ERα co-regulation. Illustration created with BioRender.com.

deSUMOylation of PELP1[16]. Previous studies have shown that PELP1 expression is significantly increased in cancer tissues[8]. PELP1 proteostasis has also been shown to be disrupted in cancer and appears to potentiate tumorigenesis[3,44]. PELP1's sub-cellular localization is altered to include the cytoplasm in many breast cancers and is thought to be an effect of increased gene/protein expression[44–46]. This localization of PELP1 was determined to promote an advanced cancer phenotype, with hallmarks being aberrant involvement in cytosolic signaling pathways (i.e., MAPK and AKT activation) that presumably promote extranuclear estrogen responses and fostering cancer cell resistance to Tamoxifen[23,25,44–46]. These observations, driven by the oncogenic overexpression of PELP1, are most likely induced by an imbalance of PELP1 protein in the cell, which likely affects the homeostasis of Rix1 complex assembly. Recent work found that depletion of PELP1 in cells induces downregulation of the other Rix1 complex members[27]. This same study also observed upregulation of PELP1 and Rix1 complex members in TNBC likely pointing to a relationship with the ribosome biogenesis pathway, which is known to be upregulated in all cancers[27,47,48]. This expression mimicry underscores the importance of cellular PELP1 levels for Rix1 complex assembly and supports PELP1's primary function being within the Rix1/Rixosome complex[14,15,18].

Finally, this brings into question the consequence of robust PELP1 overexpression observed in ERα positive breast cancers, as well as many hormonal cancers, wherein PELP1 oncogenic function may extend beyond ribosome synthesis and into SR coactivation. PELP1 overexpression in ERα positive breast cancers may exceed a threshold that disrupts Rix1 complex homeostasis, resulting in dysregulated PELP1 that is not incorporated in the Rix1 complex. This would provide a pool of PELP1 capable of adopting a structural conformation compatible with ERα activation. PELP1 functions in several cancerous hormone signaling pathways; therefore, it is likely that this mechanism of PELP1 dysregulation impacts other SRs. Future studies will be needed to uncover the PELP1 overexpression threshold effect on Rix1 complex proteostasis, and how this alters the hormone response in ERα positive and other hormonal cancer types.

## Methods
### Cell culture
HEK293FT cells (Thermo) were cultured in FreeStyle 293 Expression Media without supplementation at 37 °C, 8% CO₂, 80% humidity, 130 rpm, and were used for transient human protein expression experiments/purifications. HepG2 (ATCC) cells were maintained in Opti-MEM supplemented with 10% FBS, 1% Anti-Anti, 1% GlutaMax, and were used for dual luciferase assays. MCF-7 cells were maintained in MEM supplemented with 10% FBS, 1% Penicillin/Streptomycin, 1% Glutamine, and were used for endogenous co-immunoprecipitation experiments. Special culture conditions for E2 treatments are outlined in relevant method sections below.

### Recombinant reconstitution, isolation, and detection of human Rix1 complexes
40 mL suspension cultures of HEK293FT cells were transiently transfected with plasmid vector DNA (Supplementary Table 1) using 293fectin reagent (Thermo) and using the manufacture's protocol (1:2 DNA to transfection reagent ratio). 1 μg of total DNA was used per 1 mL of cell culture. Amounts of DNA for individual plasmids co-transfected were equal. Cells were incubated for at least 48 hours to allow for protein expression and were then harvested by centrifugation as 20 mL cell pellets. Cell pellets were stored at −80 °C until use. Each cell pellet was lysed in 1 mL lysis buffer (50 mM Hepes pH 7.3, 200 mM NaCl, 5 mM Mg₂Cl, 10% Glycerol, 0.5% NP-40, EDTA-free protease inhibitor (Roche), 1.75 ×10⁻⁴ U/mL Benzonase (Sigma)) for 30 min at

4 °C with gentle agitation on a nutator. Whole cell lysate was then clarified by centrifugation at 17k× $g$ for 35 minutes at 4 °C. The anti-FLAG M2 Affinity Gel (Sigma) was used for affinity isolation of FLAG-tagged Rix1 complex members and reconstituted complexes. Approximately 20 µL of equilibrated anti-FLAG gel was incubated with 1 mL of clarified input lysate per sample for 1 h (4 °C on a nutator). Anti-FLAG gel with bound protein was then washed with 2 mL of lysis buffer, 0.2 mL of ATP wash buffer (50 mM Hepes pH 7.3, 150 mM NaCl, 20 mM Mg₂Cl, 5% Glycerol, 5 mM ATP) 3 times with each for 5-minute incubations on ice, and finally 2 mL high salt wash buffer (50 mM Hepes pH 7.5, 600 mM NaCl, 5 mM Mg₂Cl, 5% Glycerol). 5 µL of washed and protein-bound anti-FLAG gel was used for SDS-PAGE and Western blot. Remaining 15 µL was transferred to a SigmaPrep spin column (Sigma) and used for native elution of protein using 3X FLAG peptide. 15 µL of 300 µg/mL 3X FLAG peptide (Pierce) in high salt wash buffer was incubated with 15 µL of protein-bound anti-FLAG gel for 30 minutes at 4 °C and on a nutator. Eluted protein complexes were collected by centrifugation and subject to SDS-PAGE and total protein staining (SimplySafe Stain). For Western Blot of recombinant proteins input lysate and complexes bound to anti-FLAG gel, samples were boiled at 95 °C for 10 min, loaded on an 4–15% Polyacrylamide gel, and ran at 200 V for 30 min before separated proteins were transferred onto a nitrocellulose membrane (BioRad). Membranes were blocked for 1.5 h at room temperature in 5% milk in TBST (0.1% TWEEN-20) and then incubated with primary antibody overnight at 4 °C. Primary antibody mixtures were either anti-FLAG (Sigma #7425), anti-HA (Invitrogen #26183), anti-GFP (Roche #11814460001), anti-MYC (Sigma #05-724), or anti-Tubulin (Invitrogen #MA1-80017) in 5% milk, 1% BSA in TBST. The following morning, membranes were washed three times with TBST and incubated for 1 h at room temperature with secondary antibody (anti-mouse HRP or anti-rabbit HRP, Sigma) in 5% milk, 1% BSA in TBST. Membranes were then washed three more times in TBST and then exposed using a BioRad chemiluminescent imager.

### BS3 crosslinking mass spectrometry

Purified complex (6.0 µM for PELP1 Rix1 domain – WDR18 subcomplex and 2.0 µM for Rix1 complex) was crosslinked with 1.0 mM and 1.2 mM, respectively, of bis(sulfosuccinimidyl)suberate (BS3; Sigma) in (50 mM Hepes pH 7.3, 600 mM NaCl, 5 mM MgCl₂, 5% Glycerol) at room temperature for 10 min before quenching with 300 mM Tris pH 7.5 for 15 min at 4 °C. 5 µL of the crosslinked complex was diluted to 20 µL and digested with addition of 1 µL trypsin (0.1 µg/µL – Promega) in 50 mM ammonium bicarbonate buffer overnight at 37 °C. The digests were then stored at −80 °C for subsequent MS analysis. Protein digests were analyzed by LC/MS on a Q Exactive Plus mass spectrometer (ThermoFisher Scientific) interfaced with an M-Class nanoAcquity UPLC system (Waters Corporation) equipped with a 75 µm x 150 mm HSS T3 C18 column (1.8 µm particle, Waters Corporation) and a C18 trapping column (180 µm × 20 mm) with 5 µm particle size at a flow rate of 450 nL/min. The trapping column was in-line with the analytical column and upstream of a micro-tee union which was used for venting, trapping, and as a liquid junction. Trapping was performed using the initial solvent composition. 5 µL of digested sample was injected onto the column. Peptides were eluted by using a linear gradient from 99% solvent A (0.1% formic acid in water (v/v)) and 1% solvent B (0.1% formic acid in acetonitrile (v/v)) to 40% solvent B over 70 minutes. For the mass spectrometry, a top-ten data-dependent acquisition method was employed with a dynamic exclusion time of 15 seconds and exclusion of singly charged species. The mass spectrometer was employed with a nanoflex source with a stainless-steel needle and used in the positive ion mode. Instrument parameters were as follows: sheath gas, 0; auxiliary gas, 0; sweep gas, 0; spray voltage, 2.7 kV; capillary temperature, 275 °C; S-lens, 60; scan range (m/z) of 375 to 1500; 1.6 m/z isolation window; resolution: 70,000 (MS), 17,500 (MS/MS); automated gain control (AGC), 3 ×10⁶ ions (MS), 5 ×10⁴ (MS/

MS); and a maximum IT of 100 ms (MS), 50 ms (MS/MS). Mass calibration was performed before data acquisition using the Pierce LTQ Velos Positive Ion Calibration mixture (ThermoFisher Scientific). The LC/MS raw data were first converted to an MGF format using Mascot Distiller from Matrix Science and then analyzed using the Batch-Tag Web function of the Protein Prospector web-based software developed by the UCSF Mass Spectrometry Facility. The MGF file was searched against sequences for members of the human Rix1 complex by employing the User Protein Sequence field with other search parameters, including tryptic specificity and 3 missed cleavages; precursor charge range of 2, 3, 4, and 5; monoisotopic values; parent mass tolerance of 20 ppm and fragment mass tolerance of 50 ppm; oxidation of methionine and incorrect monoisotopic assignment as a variable modifications; and in the Crosslinking field, the Link Search Type was defined as DSS. The putative crosslinked peptide output was triaged by limiting the mass error of putative crosslinks to two standard deviations from the average error (about 5 ppm); requiring a Score Difference value >4 except for the cases of intermolecular crosslinks of identical peptides or peptides less than or equal to 3 amino acid residues; and total expectation values below 1 ×10⁻⁴.

### Purification of PELP1-WDR18 and cryo-EM sample preparation

N-terminally FLAG-tagged PELP1 Rix1 domain (1-642aa) and C-terminally HA-tagged WDR18 protein was transiently co-expressed in 100 mL of HEK293FT cells using same expression protocol as above. 100 mL cell pellet was lysed using a high salt lysis buffer (600 mM NaCl, 50 mM Hepes pH 7.7, 5 mM MgCl₂, 5% Glycerol, 0.5% NP-0.4, EDTA-free protease inhibitor, 1.75 ×10⁻⁴ U/mL Benzonase) for 30 minutes at 4 °C with gentle agitation on a nutator. Whole cell lysate was then clarified by centrifugation at 17k× $g$ for 40 minutes at 4 °C. The anti-FLAG M2 Affinity Gel (Sigma) was used for affinity purification of FLAG-tagged PELP1 Rix1 domain bound to WDR18 by incubating clarified lysate with anti-FLAG gel for 1.5 h at 4 °C with gentle agitation on a nutator. Anti-FLAG gel with bound protein was washed 2 times with 10 mL high salt lysis buffer and 2 more times with 10 mL low salt buffer (150 mM NaCl, 50 mM Hepes pH 7.7, 5 mM MgCl₂). Bound protein was eluted from the anti-FLAG resin using 300 µg/mL 3X FLAG peptide in low salt buffer using two separate elutions. Each elution was incubated with anti-FLAG gel in a SigmaPrep spin column for 45 minutes at 4 °C with gentle rocking before the collection of eluted protein by centrifugation at 500 x g. Combined protein elutions were immediately used for cryo-EM sample preparation. A carbon C-flat™ 1.2/1.3 grid coated with 30 nm gold in-house was rendered hydrophilic using the Tergeo plasma cleaner (Pie Scientific). Protein solution (4 µL at 0.24 mg/ml) was deposited onto the grids and back blotted in a 90% humidity chamber at 8 °C for 3 s using an Automatic Plunge Freezer (Leica).

### Cryo-EM data collection, image processing, and model building

Images of PELP1-WDR18 were collected using a Titan Krios electron microscope (Thermo Fischer Scientific) at 300 keV with a Gatan K3 detector in super-resolution mode. Movie collection parameters are listed in Table 1. Raw movies were downsampled to a calibrated pixel size of 1.056 Å before data processing. Beam-induced motion and drift were corrected using MotionCor2[49]. Aligned and dose-weighted images were used to calculate CTF parameters using CTFFIND4[50]. Further image processing was performed in CryoSPARC v2 (Supplementary Fig. 6)[51]. Briefly, particles were template-picked and extracted from micrographs with a box size of 224 pix initially binned by 4 (4.224 Å/pix). Ab initio reconstructions were performed with curated, 2D classified particles with a bin of 2 (2.112 Å/pix). The best reconstruction representing features seen in 2D classifications was used for further map refinements with no binning (1.056 Å/pix). C2 symmetry was not applied to refinement parameters until presence of nonsymmetric features were ruled out. After applying C2 symmetry, local CTF

## Table 1 | Cryo-EM data collection, refinement, and validation statistics

| Sample | *Hs* PELP1-WDR18 |
|---|---|
| **EM data collection and processing:** | |
| Microscope | Titan Krios |
| Camera | Gatan K3 |
| Voltage (kV) | 300 |
| Magnification | 81000 x |
| Frames (no.) | 50 |
| Electron dose per frame (e⁻/Å²) | 0.8 |
| Electron dose rate (e⁻/Å²/s) | 60 |
| Calibrated pixel size (Å) | 0.528 |
| Defocus range (microns) | −1.2 - −2.2 |
| Micrographs | 2843 |
| Initial picks (no.) | 4,968,567 |
| | **PDB: 7UWF** |
| | **EMD: 26831** |
| Refined particles (no.) | 278,192 |
| Symmetry imposed | C2 |
| Global resolution (Å) | |
| FSC 0.5 (unmasked/masked) | 2.76/2.75 |
| FSC 0.143 (unmasked/masked) | 2.63/2.63 |
| FSC 0.0 (unmasked/masked) | 2.42/2.37 |
| Local resolution range (Å) | 2.3-25.5 |
| **Model refinement and validation:** | |
| Initial model | Alphafold (Q8IZL8, Q9BV38) |
| Model composition | |
| Non-hydrogen atoms | 13,174 |
| Protein residues | 1760 |
| Bonds (RMSD) | |
| Length (Å) | 0.004 |
| Angles (°) | 0.617 |
| B factors (Å²) | |
| Protein (min/max/mean) | 3.80/58.85/21.72 |
| Ramachandran plot | |
| % favored | 96.96 |
| % allowed | 3.04 |
| % outliers | 0.00 |
| Rotamer outliers (%) | 0.00 |
| MolProbity | |
| Clashscore | 6.12 |
| MolProbity score | 1.51 |
| Model-map comparison | |
| CC_mask | 0.82 |
| CC_volume | 0.80 |

refinement of particles was completed, leading to a set of tandem C2 refinements +/- on-the-fly per-particle defocus optimization. A final heterogenous C2 refinement was performed to sort out particles contributing to anisotropy, leading to a primary particle stack refined to 2.7 Å. This final particle stack and refined map was used for further C2 symmetry expansion and 3D variability analysis in cryoSPARC to assess for local motion. The cryoSPARC sharpened and local filtered map was used for molecular modeling. Half maps were used for DeepEMhancer sharpening on the highRes deep learning model, which produced the post-processed map used for figure making[52]. AlphaFold predicted models (PELP1 UniProt: Q8IZL8, WDR18 UniProt: Q9BV38) were used as starting models and fit into the cryo-EM map using rigid

body docking in Phenix[29,53,54]. Iterative rounds of real-space refinement coupled with manual building in COOT were used to improve the fit of the model[55,56]. Molprobity was used to evaluate the model (Table 1)[57]. Chimera X v1.3 was used to prepare figures[58].

### Endogenous protein co-immunoprecipitation
MCF7 cells were seeded in MEM supplemented with 10% Fetal Bovine Serum (FBS), 1% Pen/Strep and 1% L-glutamine. After 48 hours, cells were washed 5 times with Hormone Depleted medium (Phenol-Red Free MEM supplemented with 10% charcoal dextran-stripped FBS, 1% Pen/Strep, 1% L-Glutamin) and maintained 72 hours for efficient hormone depletion. 10 nM 17β-Estradiol (E2) (dissolved in 100% ethanol and stored at −20C) or 10 nM ethanol (EtOH) were then added to fresh Hormone depleted medium for 24 hours and cells were harvested. Cell pellets were resuspended in IP Lysis Buffer (25 mM Tris-HCl pH 7.5, 150 mM NaCl, 1 mM EDTA, 1% NP-40, 5% Glycerol, 1X Halt™ Protease and Phosphatase Inhibitor Cocktail (Invitrogen) and 1 mM PMSF) and incubated on ice for 20 min. After 10 min max speed centrifugation at 4 °C, supernatant was collected, 20 µg were stored and further used at Input, and 1 mg of protein extract per condition were used for preclearing using control isotype antibody (Millipore #12-370). Protein extracts were then incubated overnight with control isotype antibody, PELP1 antibody (Bethyl Labs #A300-180A-M), or WDR18 antibody (Sigma #HPA050200). The day after, 20 µL of Dynabeads Protein G (Invitrogen #10004D) were added and incubated 2 hours at 4 °C on a wheel. Beads were placed on a magnetic rack and supernatant was collected as Flow Through fraction. Beads were further washed 4 times with IP Lysis Buffer containing 0.1% Triton X-100 and eluted in 2X Laemmli containing 150 mM DTT. Input, FT (flow through) and IP fractions were boiled at 95 °C for 15 min and loaded on an 8% Acrylamide gel and transferred onto nitrocellulose membrane (BioRad) for Western Blot. Membranes were incubated overnight at 4 °C with ERα antibody (1/500, Millipore #06-935), PELP1 antibody (1/5000, Bethyl Labs #A300-180A-M) or WDR18 antibody (1/125, Sigma #HPA050200) and imaged using Azur Biosystem c600 imager.

### HepG2 and MCF7 3x-ERE dual luciferase assays and statistics
HepG2 and MCF7 cells were used for luciferase assay experiments and cultured in phenol red-free Opti-MEM supplemented with 10% charcoal-dextran stripped FBS (GeminiBio), 1% Anti-Anti, 1 % GlutaMax (MCF7 cells were hormone depleted in this media for 48 hours before beginning the experiment). Cells in 24-well plates at ~60% confluency were transfected with luciferase assay and protein expression plasmids using Lipofectamine 3000 (Invitrogen) per manufacturer's instructions. Each cell condition was performed in triplicate (HepG2) or quadruplet (MCF7). For HepG2 assays, plasmid DNA contents per well included 50 ng pTK encoding Renilla control luciferase, 100 ng pGL3-TATA-3xERE encoding Firefly luciferase, 50 ng ERα, 250 ng experimental expression plasmid 1, 250 ng experimental expression plasmid 2. For MCF7 assays, plasmid DNA contents per well included 100 ng pTK encoding Renilla control luciferase, 200 ng pGL3-TATA-3xERE encoding Firefly luciferase, 500 ng experimental expression plasmid 1, 500 ng experimental expression plasmid 2 (Supplementary Table 1). The same amount of expression plasmid for experimental conditions was used even when assessed alone. In these conditions, to maintain equal amounts of DNA transfected in all wells, the remaining 250 ng (HepG2) or 500 ng (MCF7) of DNA was filled in using pcDNA3.1 empty vector. The DNA transfection mixture was incubated with cells at 37 °C for 8 hrs, removed from the cells, washed once with phenol red-free Opti-MEM, and then incubated with fresh media overnight. The following day, cells were treated with 10 nM E2 or EtOH for at least 18 h at 37 °C and then Luciferase activity was assessed by following the Promega dual luciferase assay kit instructions and using a ClairoStar automated plate reader for luminescence. All data were normalized to Renilla luciferase activity as a transfection control. Significance

between select experimental conditions was determined by first assessing for variance using Single Factor ANOVA, followed by two sample $t$-Tests assuming unequal variances.

## Reporting summary

Further information on research design is available in the Nature Portfolio Reporting Summary linked to this article.

## Data availability

The cryo-EM map and atomic coordinates for PELP1-WDR18 have been deposited in the Electron Microscopy Data Bank and PDB under the following access numbers: EMD-26831 and 7UWF. The mass spectrometry data has been deposited to the ProteomeXchange under project name: Chemical crosslinking and mass spectrometry of the human Rix1 complex, with the accession number PXD037729. All other data are present in the manuscript, supplementary information and supplementary data files. Source data are provided as a Source Data file. Plasmids created and used in this study will be made available upon request. Source data are provided with this paper.

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

## Acknowledgements

We would like to thank the Molecular Microscopy Consortium at the NIEHS for their help with cryo-EM data collection and processing, along with the Mass Spectrometry Research and Support Group at the NIEHS for their help with mass spectrometry data collection and analysis. We would like to thank Dr. Rick Huang and Allison Zeher for their help with cryo-EM data collection at the NCI/NICE Cryo-EM Facility. We would like to thank Dr. Yin Li at NIEHS for donation of luciferase assay reagents and Dr. Robert Petrovich from the NIEHS Protein Expression Core Facility for help with mammalian cell culture. We would also like to thank Dr. Brad Klemm and Dr. Yukitomo Arao for their critical reading of the manuscript. This work was supported by the US National Institutes of Health Intramural Research Program; US National Institute of Environmental Health Sciences (NIEHS) (ZIA ES103247 to R.E.S.,1ZI CES102488 to J.G.W., 1ZI CES103206 to L.J.D., ZIC ES103326 to M.J.B, and ZIA ES103331 to J.R.). Work in the A.J.W. lab is supported by Blood Cancer UK (21002), the UK Medical Research Council (MR/T012412/1), the Kay Kendal Leukaemia Fund and the European Cooperation in Science and Technology (COST) Action CA18233. We would also like to acknowledge the NIH-Oxford-Cambridge Scholars Program for support of J.G. as a graduate student in the labs of A.J.W. and R.E.S.

## Author contributions

R.E.S., A.J.W., and J.G., conceived and designed the study. J.G., E.G.V., J.M.K., M.J.B, and R.E.S. performed cryo-EM analysis, structure determination, and structure refinement. J.G. purified Rix1 complexes and carried out Co-IP experiments. J.G.W. and L.J.D. processed and analyzed mass spectrometry data. J.R, F.L.C., and J.G. carried out experiments addressing nuclear receptor binding and activation. J.G. and R.E.S. wrote and revised the manuscript which was edited and approved by all the authors.

## Funding

## Competing interests

The authors declare no competing interests.
