## [Peer Review File · Nature Communications]

Cryo-EM Reveals the Architecture of the PELP1-WDR18 Molecular Scaffold

Editorial Note: Parts of this Peer Review File have been redacted as indicated to remove third party material where no permission to publish were obtainedREVIEWER COMMENTS

Reviewer #1 (Remarks to the Author):

In this very interesting manuscript Gordon and colleagues investigate the architecture of the PELP1 molecular scaffold within the human Rix1 complex. PELP1 was originally identified and characterized as a transcriptional coactivator of steroid hormone receptors, including ERα. PELP1 is overexpressed in 60-80% of breast cancers, where its level directly correlates with tumor grade, metastasis, and endocrine therapy resistance. Later studies have surprisingly revealed that PELP1 plays an essential role in ribosome biogenesis by serving as the core component of the Rix1 complex. How PELP1 coordinates its diverse scaffolding function as a steroid hormone receptor coactivator and component of the Rix1 complex has been puzzling. A highly significant question that awaited to be answered is whether PELP1 scaffolds the estrogen receptor to the Rix1 complex to promote breast tumorigenesis. To answer this question the authors, provide a cryo-EM structure of the human PELP1 Rix1 domain bound to WDR18 with visualization PELP1's several LxxLL motifs that have been implicated in ERα co-activation. With supporting cellular experiments, they provide a model in which the PELP1-WDR18 assembly, presumably within the Rix1 complex, would prevent ERα binding and coactivation. This work is likely to have an impactful contribution to the literature by providing fundamental insights into the diverse function of PELP1 and the assembly of complex biomolecules, which potentially have broad relevance not only to the field of ribosome biogenesis but also cancer biology. However, there are a few concerns denoted below that the authors should address to solidify their conclusion and increase the broad significance of their work.

1. It is not clear why the cryo-EM structure was determined only for the PELP1 Rix1 domain-WDR18 sub-complex, and not full-length PELP1 assembled with WDR18, TEX10 and SENP3. Although the C-terminus of PELP1 is predicted to be unstable, the authors demonstrate that reconstitution with the other Rix1 complex components (particularly SENP3) seems to stabilize the region (less C-terminus degradation). The information gained with full-length PELP1 would be much more valuable and would increase the significance of the work.

2. The BS3 crosslinking mass spectrometry experiments show different inter and intra-molecular cross-links depending on whether the full-length or Rix1 domain of PELP1 was used. The authors do not discuss this. There are a lot more intra- and inter-molecular PELP1-PELP1 cross-links when only the PELP1 Rix1 domain is expressed. The full-length PELP1 shows cross-links with TEX10 and SENP3 on the N-terminus, which is not reflected in the author's interaction model. This suggests that the model created with only the Rix1 domain may not be physiologically accurate. It would be important to reconcile these differences, particularly with the cryo-EM data.

3. Fig 6A shows that there is a substantial amount of PELP1 that is not in complex with WDR18 (according to the Input and FT lanes). The IP conditions that were used may not have been adequate for extracting PELP1 or ER bound to chromatin, which is likely where the liganded-ER-PELP1 interaction would occur. It would be important to determine whether the lysis buffer conditions used only allow to look at nuclear interactions and not chromatin or nucleolar interactions. Additionally, is overexpression of WDR18 sufficient to sequester away most of endogenous PELP1?

4. Does PELP1 exist in different sub-complexes such as PELP1-WDR18 or PELP1-TEX10-SEN3 at endogenous levels? For example, would a PELP1-TEX10-SEN3 sub-complex interact with ERa? This would be important to determine, specifically because the data presented suggests that the Rix1 domain of PELP1 likely behaves differently when not bound to WDR18.

5. Fig6B: It would be important to provide WB to show levels of ectopically expressed PELP1 and WDR18. Based on Fig1B, Fig4H, SupFig2, the levels of FL PELP1 are always lower when co-expressed with FL WDR18. Fig4H shows that PELP1 levels are not reduced as much when co-expressed with WDR18 1-344. These differences in PELP1 levels could be an alternative explanation for the data. The addition of similar experiments in a more physiologically relevant system (MCF7 cells and endogenous qRT-PCR of ERa targets shown to be regulated by PELP1) would help solidify the conclusions.

Minor points:

1. Blots with higher exposures should be provided for Fig4H. The amount of PELP1 co-IPed with WDR18 is not clearly visible.

2. It should be Renilla and not Ranilla luciferase.

Reviewer #2 (Remarks to the Author):

The manuscript by Gordon and colleagues provides the first cryo-EM structure containing PELP1, a scaffolding protein involved in transcription factor co-activation, heterochromatin maintenance and ribosome biogenesis. The authors reconstituted members of the Rix1 complex in 293T cells and found that the N-terminal domain of PELP1 and WDR18 form a stable sub-complex. Cryo-EM with PELP1 and WDR18 suggests that these proteins form a tetramer containing 2 PELP1 and 2 WDR18 subunits. Importantly, this structure resembles the orthologous yeast Rix1-Ipi3 complex. Interestingly, 7 of the 11 LXXLL domains of PELP1 are localized on the internal face of the PELP1-WDR18 complex, suggesting that these domains are not positioned for interacting with steroid receptors (SRs). The authors use luciferase assays to show that expression of WDR18 inhibits E2-induced ERE-luciferase activity. Overall these studies are very important for the understanding of the RIX1 complex and also the ability of PELP1 to function as a coactivator protein, but there are several points that should be addressed.

The positioning/availability of the PELP1 LXXLL motifs is very interesting. Many publications suggest that PELP1 acts as a co-activator for SRs and other transcription factors as well. Is the presence of WDR18 promoting the internalized LXXLL confirmation? Does PELP1 form a dimer with a similar confirmation alone? It is important to note that the Cryo-EM structure is only the N-terminal half of PELP1, does the presence of the C-terminal half of PELP1 alter the PELP1-WDR18 cryo-EM structure.

The authors suggest that WDR18 may act as an inhibitor of PELP1 co-activation, but luciferase assays in overexpression systems are insufficient to make this conclusion. WDR18 knockdown, could support this claim. In addition, endogenous E2-driven gene expression could further support this conclusion, but as is these conclusions are an overinterpretation of the presented data.

Suggesting that therapeutically targeting PELP1 oncogenic functions through expression WDR18 is not supported by the studies presented.

Reviewer #3 (Remarks to the Author):

In the manuscript by Gordon et al., entitled, "Architecture of the PELP1 Molecular Scaffold Reveals Insights into Nuclear Receptor Coactivation" the authors characterized the structure at high resolution of the subcomplex formed by PELP1 Rix1 N-terminal domain and WDR18 WD domain. Binding and functional assays were used to characterize the consequence of complex formation of PELP1-WDR18 on estrogen receptor coactivation.

Major comments:

-Cellular localization of PELP1 should be discussed. Where are PELP1-WDR18 and PELP1-ER complex formed? The phosphorylation of PELP1 has been shown to be important for nuclear receptor coactivation and should be discussed.

-Is PELP1 a primary or secondary nuclear receptor coactivator for ER?

-It is not clear that oligomeric state observed in the cryoEM structure of the subcomplex correspond to the oligomeric state of the full-length complex. It should be clarified.

-The authors used crosslink mass spectrometry to characterize human Rix1 complex but these data are not discussed in context of the cryoEM structure of the subcomplex.

-The authors used the predicted alphafold models for the PELP1 Rix1 domain as an initial model for cryoEM refinement. The comparison of the final model of the complex with the alpha-fold model of the isolated protein should be discussed. This is important as the authors show that the LXXLL motifs are important for the stability of the domain and is unlikely, without conformational change, to be involved in nuclear receptor interaction. Can other region outside of the LXXLL motifs be involved in ER interaction?

-LXXLL motifs interact with nuclear receptors through non polar interactions but also formed electrostatic interactions through a charge clamp. What about sequence conservation of the various LXXLL motifs and surrounding residues?

Reviewer #4 (Remarks to the Author):

The manuscript authored by Gordon J and entitled "Architecture of the PELP1 molecular scaffold reveals insights into nuclear receptor coactivation" describes the purification and structural characterization of a core hertetrameric Rix complex composed of two PELP1 and two WDR18 molecules. The cryo-EM structure shows that the eleven putative steroid hormone receptor binding motifs LxxLL present in each PELP1 molecule are either not solvent accessible or engaged in interactions that prevent SR binding. This finding provides anew mechanism for regulating SR interactions with the PELP1 co-activator. The structural results are sustained by in vivo colP experiments showing that the Eostrogen receptor (ER) does not interact with the PELP1-WDR18 sub-complex. The structural results are of good quality but

match an already published structure of the homologous yeast complex which in addition contains the IPI1/TEX10 subunit. The major new finding lays therefore in the positioning of the putative SR binding motifs which are absent in yeast.

The following issues need to be addressed:

1- P4: The initial description of the reconstituted RIX complex should mention the fact that PELP1 is strongly proteolyzed thus leaving only 20-30% of the intact full-length protein. This proportion is reduced when all 4 proteins are overexpressed but still leaves a mixture of 50% intact and 50% of cleaved PELP1. From the band intensities on the Coomassie stained gel the SENP3 and TEX10 subunits appear underrepresented. Therefore, the complex appears to be a mixture (described in the second section) and the authors cannot conclude at the end of section 1 that the complex is stoichiometric

2- A C-terminal FLAG tagged PELP1 brings intact Rix1 complex which appears more relevant and homogeneous. The authors should indicate why was this complex not analysed in cryo-EM as it might have provided a more detailed view of the functional complex.

3- Concerning the cross-linking mass spectrometry analysis, I was concerned about Supplemental figure 3 showing that only minute amounts of the complex is actually cross-linked. Does this affect the number and amounts of detectable cross-links? Secondly, I found that the cross-links between PELP1 and WDR16 are different in the Rix1 complex (supplemental fig3) than in the PELP1-WDR16 sub complex (supplemental 4). Additional cross-links are found for the PELP1-WDR16 sub complex as well as additional intralinks within PEPL1. This could mean that the structure of the subcomplex is different than that of the full Rix1 complex. The authors should comment these differences.

4- The title is misleading. This work addresses the structure of PEPL1 in a Rix1-sub complex which does not recapitulate the structure of free PELP1 which exerts the co-activator function. The structure of this very important Steroid Receptor co-activator can unfortunately not be determined (yet) in its functional state because of intrinsic flexibility without association with WRD18.

5- Based on the solvent accessibility and the availability of key Leucine residues, the authors conclude that the 11 LxxLL motifs are not compatible with steroid receptor binding. To support this conclusion, they provide Colp studies using an anti WRD18 antibody to show that ER α does not interact with WDR18 containing complexes (Fig.6a). However, control experiments shown in supplemental Fig 9 show that ER α interacts very weakly with PEPL1-containing complexes and the weak interaction can only be detected in a strongly overloaded lane. Therefore, only minute amounts of ER α are bound to free PELP which weakens the conclusions. I recommend that the authors perform binding experiments with purified steroid receptors.

6- In the discussion the authors claim “Interestingly, we observed the N-terminal PELP1 LxxLL motifs LM1 and LM2 sampling ordered and disordered states upon 3D variability analysis of our cryo-EM data”. Unless I overlooked it, this data is not provided.

7- The authors provide an interesting discussion about the role of PELP1 overexpression in cancer cells which, to my opinion, included much speculation especially since the Rix1 complex has multiple additional roles in ribosome assembly and heterochromatin maintenance that could interfere with the oncogenic phenotype. It is also unknown what regulates the interactions between PELP1 and WDR18 is it only protein expression levels or can additional mechanisms such as post-translational modification be involved in addition.

Minor comments

P5 PELP1 has a unique biochemical composition, It should read a unique amino acid composition

P6: The authors should mention in the main text that complex on which the XL-MS experiments were performed contains the C-terminally tagged version of PELP1.

P7 avoid repeats in the sentence “Single-particle reconstruction resulted in a 2.7 Å resolution reconstruction”

Legend figure 2 based ON instead of based OFF

Figure 2 it is difficult to relate PELP1 structure in Fig 2f to the overall structure in Fig2e. there is a red colored helix in 2d which is not found in 2f, labeled

The authors should reference the supplemental figures 5-7 in the text

Figure 4e, correct “blade contact with PELP1?”

P12 mediated instead of mediated

Rebuttal NCOMMS-22-17303

Reviewer Comments (*black italics*) and our response (*blue*)

Reviewer #1 (Remarks to the Author):

In this very interesting manuscript Gordon and colleagues investigate the architecture of the PELP1 molecular scaffold within the human Rix1 complex. PELP1 was originally identified and characterized as a transcriptional coactivator of steroid hormone receptors, including ERα. PELP1 is overexpressed in 60-80% of breast cancers, where its level directly correlates with tumor grade, metastasis, and endocrine therapy resistance. Later studies have surprisingly revealed that PELP1 plays an essential role in ribosome biogenesis by serving as the core component of the Rix1 complex. How PELP1 coordinates its diverse scaffolding function as a steroid hormone receptor coactivator and component of the Rix1 complex has been puzzling. A highly significant question that awaited to be answered is whether PELP1 scaffolds the estrogen receptor to the Rix1 complex to promote breast tumorigenesis. To answer this question the authors, provide a cryo-EM structure of the human PELP1 Rix1 domain bound to WDR18 with visualization PELP1's several LxxLL motifs that have been implicated in ERα co-activation. With supporting cellular experiments, they provide a model in which the PELP1-WDR18 assembly, presumably within the Rix1 complex, would prevent ERα binding and coactivation. This work is likely to have an impactful contribution to the literature by providing fundamental insights into the diverse function of PELP1 and the assembly of complex biomolecules, which potentially have broad relevance not only to the field of ribosome biogenesis but also cancer biology. However, there are a few concerns denoted below that the authors should address to solidify their conclusion and increase the broad significance of their work.

We thank the reviewer for their supportive comments and suggestions for improvement.

1. It is not clear why the cryo-EM structure was determined only for the PELP1 Rix1 domain-WDR18 sub-complex, and not full-length PELP1 assembled with WDR18, TEX10 and SENP3. Although the C-terminus of PELP1 is predicted to be unstable, the authors demonstrate that reconstitution with the other Rix1 complex components (particularly SENP3) seems to stabilize the region (less C-terminus degradation). The information gained with full-length PELP1 would be much more valuable and would increase the significance of the work.

We completely agree that solving the structure of the entire Rix1 complex would be valuable and significant. This was our original goal when we initiated this project. Unfortunately, despite our best efforts we have not been able to get the entire Rix1 complex to behave on cryo-EM grids. We have made grids of the entire complex but the PELP1-WDR18 subcomplex remains the dominant view on the micrographs and from the 2D classes. This suggests that either the complex falls apart at the air-water interface or the C-terminal region of PELP1 and its associated factors (TEX10 and SENP3) is very flexible and dynamic making it invisible by cryo-EM.

The C-terminus of PELP1 is not predicted to have any structure and it also has a very unusual amino acid composition, while TEX10 and SENP3 do appear to stabilize parts of the C-terminus this is not sufficient to enable structure determination of the full complex. Given that overcoming the inherent flexibility of the C-terminus is a huge technical challenge we feel that solving the structure of the full complex goes beyond the scope of the current work. Moreover, given the large size of the C-terminus it's highly possible that the C-terminus functions as a flexible tether between the PELP1/WDR18 region and the TEX10-SENP3 interacting region, which would make structure determination of the full complex not possible unless TEX10-SENP3 stably anchor onto the structured PELP1/WDR18 region. We do detect a few crosslinks between TEX10 and SENP3 with the PELP1/WDR18 subcomplex, however these are primarily mediated by unstructured regions of TEX10 and SENP3.

While structure determination of the full complex is not yet possible, we performed additional IP experiments to probe the TEX10-PELP1 and SENP3-PELP1 interfaces (Please see revised Figure 1). Through these additional experiments we learned that SENP3 can associate with the isolated C-terminal domain. In contrast we were surprised to discover that TEX10 requires both the N- and C-terminal

domains of PELP1 for binding. These new results led us to update the model of the full Rix1 complex shown in Figure 7.

2. The BS3 crosslinking mass spectrometry experiments show different inter and intra-molecular cross-links depending on whether the full-length or Rix1 domain of PELP1 was used. The authors do not discuss this. There are a lot more intra- and inter-molecular PELP1-PELP1 cross-links when only the PELP1 Rix1 domain is expressed. The full-length PELP1 shows cross-links with TEX10 and SENP3 on the N-terminus, which is not reflected in the author's interaction model. This suggests that the model created with only the Rix1 domain may not be physiologically accurate. It would be important to reconcile these differences, particularly with the cryo-EM data.

Based on the comments from this reviewer and several other reviewers we realized that the section on BS3 crosslinking along with the original figures were confusing and hard to follow. Thus, we have re-written this entire section of the manuscript and prepared new figures to help make the major points clear. The reviewer is correct that we see many more PELP1-PELP1 crosslinks with the WDR18 subcomplex than with the entire Rix1 complex. This is because of the abundance of the sample. The Rix1 domain-WDR18 subcomplex is very stable and can be purified to much higher yields than the entire Rix1 complex. The higher concentration leads to a significant increase in the number of high confidence crosslinks detected during mass spec analysis.

In our original figures we included crosslinks from the epitope tags, which are very flexible and are not likely to be physiologically relevant. To avoid confusion, we have removed all crosslinks arising from the epitope tags from the figure, however this information is still available in the supplemental data file. To make it easier to compare PELP1-WDR18 crosslinks from the subcomplex with those from the Rix1 complex we have added an inset in Supplementary Fig. S4b directly comparing the two. This figure illustrates that there is excellent agreement in the pattern of crosslinks from the two different samples, supporting that the sub-complex is reflective of the WDR18-PELP1 structure within the full Rix1 complex. Furthermore, the cross-links between PELP1-WDR18 in both samples (along with the cryo-EM structure) are in excellent agreement with the structure of Rix1-Ipi3-Ipi1 bound to the yeast pre-60S ribosome. This supports that the isolated PELP1/WDR18 model is physiologically relevant, with one exception. We fully anticipate that analogous to the yeast complex, PELP1/WDR18 will lose symmetry when ribosome bound, leading to minor conformational changes along the axis of symmetry.

We were not able to detect any crosslinks arising from the C-terminus of PELP1 but we believe this is a limitation of using an amine reactive crosslinker and the unusual amino acid composition of the PELP1 C-terminus. As mentioned by the reviewer we do detect a few crosslinks between the N-terminus of PELP1 and TEX10 and SENP3 that contrasts with our model suggesting that it is the C-terminus of PELP1 that mediates these interactions. The crosslinks from SENP3 and the N-terminus of PELP1 are all through the unstructured N-terminal half of SENP3. These crosslinks simply indicate that the flexible N-terminus of SENP3 can be found near the PELP1-Rix1 domain, which is not unexpected given that these proteins are in the same complex together. During the revision we performed additional IP experiments to better map the interfaces within the complex and discovered that TEX10 requires both the N- and C-terminal domains of PELP1 for binding. Thus, we have updated our model shown in Figure 6 to reflect this.

3. Fig 6A shows that there is a substantial amount of PELP1 that is not in complex with WDR18 (according to the Input and FT lanes). The IP conditions that were used may not have been adequate for extracting PELP1 or ER bound to chromatin, which is likely where the liganded-ER-PELP1 interaction would occur. It would be important to determine whether the lysis buffer conditions used only allow to look at nuclear interactions and not chromatin or nucleolar interactions. Additionally, is overexpression of WDR18 sufficient to sequester away most of endogenous PELP1?

We thank the reviewer for the suggestion to try additional IP conditions. There are several published reports showing an IP interaction between ERalpha and PELP1 (Vadlamudi et al JBC, 2001, Barletta et al Mol. Endocrinol, 2004, Brann et al Mol Cell Endocrinol, 2008, Raj et al eLife, 2017).

As suggested by the reviewer, we repeated the IP using ChIP in MCF7 cells. We crosslinked the sample, performed a nuclear fractionation followed by chromatin extraction and then IP with the antibody for

ERalpha. Through this approach we enrich for chromatin bound ERalpha, however we still struggled to detect the presence of endogenous PELP1. The blot from this experiment is shown below but we think this result further reiterates that PELP1 is likely a secondary or even tertiary interaction partner of ERalpha. This interaction may also be transient, dynamic, cell-type specific, and/or very sensitive to the antibodies used for immunoprecipitation.

Jason Carroll's lab performed a quantitative multiplexed method (qPLEX-RIME1) to delineate the dynamics of the ERalpha interactome in steady state breast cancer cells (Papachristou et al Nat. Commun. 2018). Using this highly sensitive approach numerous ERalpha interacting proteins were identified. While PELP1 was detected through this sensitive method it was not identified as one of the 300 most frequently enriched ERalpha interactors in MCF7 cells. The results from this manuscript support our observations that it is difficult to detect a robust interaction between PELP1 and ERalpha, reinforcing that PELP1 is likely not a primary interaction partner of ERalpha.

ERalpha CHIP from MCF7 cells performed in the presence and absence of E2. ERalpha was retained but we could not detect PELP1 binding to ERalpha under these conditions.

4. Does PELP1 exist in different sub-complexes such as PELP1-WDR18 or PELP1-TEX10-SEN3 at endogenous levels? For example, would a PELP1-TEX10-SEN3 sub-complex interact with ERα? This would be important to determine, specifically because the data presented suggests that the Rix1 domain of PELP1 likely behaves differently when not bound to WDR18.

Given the large number of PELP1 interaction partners that have been identified (reviewed in Sareddy and Vadlamudi Gene, 2016 and Gonugunta et al Endoc Relat Cancer, 2014) we think it's highly likely that PELP1 exists in many different subcomplexes within the cell. The isolated C-terminal half of PELP1 can bind to SENP3 independent of WDR18 (data now shown in Fig. 1), but in contrast TEX10 requires both the N and C-terminal domains of PELP1 for binding. A recently published genomic and proteomics analysis (Lui et al Cancers, 2022) revealed that PELP1 knockout leads to the downregulation of all the components of the rixosome including WDR18, TEX10, SENP3, LAS1L, and NOL9, suggesting that there is a strong co-dependence on protein levels/stability between rixosome complex members with one another. From overexpression followed by IP of PELP1 alone we can detect endogenous TEX10, WDR18, and SENP3 by Coomassie Staining (please see Supplementary Fig. 2). While we also observe the presence of other minor bands, TEX10, WDR18, and SENP3 are the most prominent suggesting that PELP1's primary interaction partners are the members of the rixosome, at least in HEK293 cells, which it is important to note do not express ERalpha.

Heatmap showing the downregulation of TEX10, WDR18, NOL9, LAS1L, and SENP3 following the knockout of PELP1 in MBA-MB-231 cells. This image is from Lui et al Cancers, 2022.

[REDACTED]

5. Fig6B: It would be important to provide WB to show levels of ectopically expressed PELP1 and WDR18. Based on Fig1B, Fig4H, SupFig2, the levels of FL PELP1 are always lower when co-expressed with FL WDR18. Fig4H shows that PELP1 levels are not reduced as much when co-expressed with WDR18 1-344. These differences in PELP1 levels could be an alternative explanation for the data. The addition of similar experiments in a more physiologically relevant system (MCF7 cells and endogenous qRT-PCR of ERα targets shown to be regulated by PELP1) would help solidify the conclusions.

We thank the reviewer for suggesting these important additional experiments to solidify our conclusions. As suggested by the reviewer we carried out western blots for the ectopically expressed PELP1 and WDR18 from an experimental repeat of the HepG2 reporter assay in Fig. 6b. These blots are now included in Supplementary Fig. 11.

We also repeated the ERα reporter assay in MCF7 cells, in which we relied on the expression of endogenous ERα in the presence and absence of E2. While the E2 responsive signal is not as high in the MCF7 cells, likely because we are not overexpressing ERα, the trends we see mimic those from the HepG2 cells. We see activation when PELP1 is over-expressed alone, but this activation is decreased with the addition of WDR18. The results from this are now included as an additional panel in Fig. 6c.

We appreciate the suggestion to use qRT-PCR to look at the expression of ERα targets regulated by PELP1. Unfortunately to the best of our knowledge these gene targets have not been well characterized. We looked in the literature to identify putative PELP1-ERα targets. Based on a transcriptome analysis following PELP1 si-RNA knockdown in ZR75 cells only about 12% of PELP1 targets are predicted to be Estrogen responsive (Mann et al Molecular Oncology, 2013; <https://doi.org/10.1016/j.molonc.2013.12.012>). We selected 3 of these genes (RARα, COL18A1, COL12A1) for RT-PCR analysis in MCF7 cells but were unable to detect a significant estrogen response following treatment with E2, which prevented us from looking at the impacts of PELP1 or WDR18 overexpression or knockdown. In contrast we can detect induction with the well characterized ERα target TFF3. ER activating complexes formed within the cell are likely to be very heterogenous and PELP1 may only be a part of a small percentage of these activating complexes, making detection and analysis of in vivo targets very challenging. It is also possible that PELP1-ERα target genes elicit a dynamic response to hormone exposure making it difficult to find the correct window of time in which to perform assays. ERα co-activation is a complex and dynamic process and there is still much to be learned about how PELP1 functions as a co-regulator of ERα.

qRT-PCR of selected genes from MCF7 cells in the presence and absence of E2.

Minor points:

1. Blots with higher exposures should be provided for Fig4H. The amount of PELP1 co-IPed with WDR18 is not clearly visible.

To better view the PELP1 bands, an image of this total protein stained SDS-PAGE gel that has been adjusted with higher contrast is now provided in Fig. 4h.

2. It should be *Renilla* and not *Ranilla* luciferase.

Thank you catching this typo. It has been corrected.

Reviewer #2 (Remarks to the Author):

The manuscript by Gordon and colleagues provides the first cryo-EM structure containing PELP1, a scaffolding protein involved in transcription factor co-activation, heterochromatin maintenance and ribosome biogenesis. The authors reconstituted members of the Rix1 complex in 293T cells and found that the N-terminal domain of PELP1 and WDR18 form a stable sub-complex. Cyro-EM with PELP1 and WDR18 suggests that these proteins form a tetramer containing 2 PELP1 and 2 WDR18 subunits. Importantly, this structure resembles the orthologous yeast Rix1-Ipi3 complex. Interestingly, 7 of the 11 LXXLL domains of PELP1 are localized on the internal face of the PELP1-WDR18 complex, suggesting that these domains are not positioned for interacting with steroid receptors (SRs). The authors use luciferase assays to show that expression of WDR18 inhibits E2-induced ERE-luciferase activity. Overall these studies are very important for the understanding of the RIX1 complex and also the ability of PELP1 to function as a coactivator protein, but there are several point that should be addressed.

We thank the reviewer for commenting on the importance of our work and valuable suggestions for improvement.

The positioning/availability of the PELP1 LXXLL motifs is very interesting. Many publications suggest that PELP1 acts as a co-activator for SRs and other transcriptions factors as well. Is the presence of WDR18 promoting the internalized LXXLL confirmation? Does PELP1 form a dimer with a similar confirmation alone? It is important to note that the Cryo-EM structure is only the N-terminal half of PELP1, does the presence of the C-terminal half of PELP1 alter the PELP1-WDR18 cryo-EM structure.

We agree with the reviewer that the positioning of the LXXLL motifs is interesting and not what we would have predicted before solving the structure. We think that it is a combination of PELP1 dimerization and binding of WDR18 that promotes the inaccessible LXXLL conformations. There is a large PELP1-PELP1 interface suggesting that it could form a dimer on its own, however based on the structure we think WDR18 functions as a tether to hold the two PELP1 protomers in place. The C-terminal half of PELP1 is predicted to be completely unstructured, thus we do not think that the C-terminal half on its own or when bound to TEX10 and SENP3 will alter the PELP1-WDR18 structure. Moreover, our BS3 crosslinking analysis supports that the PELP1-WDR18 interfaces remain the same in the presence and absence of the rest of the Rix1 complex. While we anticipate that the PELP1-WDR18 interfaces are maintained in the full Rix1 complex we assume that analogous to the Yeast Rix1 complex, the PELP1-WDR18 region within the complex will become asymmetric when binding to TEX10 and the ribosome.

The authors suggest that WDR18 may act as an inhibitor of PELP1 co-activation, but luciferase assays in overexpression systems are insufficient to make this conclusion. WDR18 knockdown, could support this claim. In addition, endogenous E2-driven gene expression could further support this conclusion, but as is these conclusion are an overinterpretation of the presented data.

To provide additional support for this model with have added several additional experiments to our revised manuscript. First, we repeated the luciferase assays in MCF7 cells relying on endogenous ER. While the signal is less robust, we observe the same trends that PELP1 alone, but not in complex with WDR18, can activate ER. As mentioned above, following a similar comment from Reviewer #1 we also tried to look at the expression of PELP1 mediated E2-driven targets by qRT-PCR. Unfortunately, to the best of our knowledge these targets have not been well characterized. We selected a few putative PELP1-ERalpha targets to test but could not observe any E2 activation.

Suggesting that therapeutically targeting PELP1 oncogenic functions through expression WDR18 is not supported by the studies presented.

Following the suggestion of the reviewer we have removed this comment from the discussion.

Reviewer #3 (Remarks to the Author):

In the manuscript by Gordon et al., entitled, "Architecture of the PELP1 Molecular Scaffold Reveals Insights into Nuclear Receptor Coactivation" the authors characterized the structure at high resolution of the subcomplex formed by PELP1 Rix1 N-terminal domain and WDR18 WD domain. Binding and functional assays were used to characterize the consequence of complex formation of PELP1-WDR18 on estrogen receptor coactivation.

Major comments:

-Cellular localization of PELP1 should be discussed. Where are PELP1-WDR18 and PELP1-ER complex formed? The phosphorylation of PELP1 has been shown to be important for nuclear receptor coactivation and should be discussed.

We thank the reviewer for these suggestions, and we have added this relevant information to the discussion section of the manuscript. Work from Castle et al (Mol Bio Cell, 2012) showed that PELP1 and WDR18 are primarily nuclear/nucleolar proteins. However, work from Girard et al (JBC, 2017) and several others (Kumar et al 2009, Vadlamudi et al 2005) has shown that PELP1 is found in the cytoplasm in many invasive breast cancers. Vadlamudi et al, JBC, 2001 demonstrated that PELP1 and ERalpha associate with one another in E2 treated nuclear extracts.

-Is PELP1 a primary or secondary nuclear receptor coactivator for ER?

This is discussed in more detail above but based on the existing literature we assume that PELP1 is a secondary nuclear receptor coactivator for ER. Whole genome analysis of the PELP1 transcriptome in ZR75 breast cancer cells revealed that only about 12% of PELP1 regulated genes are established estrogen responsive genes (Mann et al Molecular Oncology, 2013).

-It is not clear that oligomeric state observed in the cryoEM structure of the subcomplex correspond to the oligomeric state of the full-length complex. It should be clarified.

Based on the structure of the ribosome bound yeast Rix1 complex, we assume that the 2:2 stoichiometry of PELP1 and WDR18 exists in both the subcomplex and the full complex. Unfortunately, we do not know the stoichiometry of TEX10 and SENP3. In the yeast complex, there is one copy of TEX10 (Ipi1) bound to the PELP1-WDR18 tetramer, thus this stoichiometry may be conserved in higher organisms, however yeast Ipi1 and human TEX10 are quite a bit different from one another. Human Tex10 is about 3 times the size of yeast Ipi1 and it is predicted to form a large alpha-helical structure (similar to the PELP1 Rix1 domain). Only the first ~300 residues of Tex10 share sequence homology with yeast Ipi1. A small loop from the C-terminus (residues 245-268) of Ipi1 was shown to bind along an interface between Rix1 (Pelp1) and Ipi1 (WDR18), however this region of Ipi1 is not well conserved in Tex10, so it remains unclear if this binding interface is the same in the human complex. During the revision we performed additional IP experiments that revealed the C-terminus of PELP1 is sufficient for SENP3 binding but that TEX10 requires both the N- and C-terminus of PELP1 for binding. TEX10's reliance on both domains suggests it requires oligomerization of PELP1 and/or the presence of WDR18 for Rix1 complex formation.

-The authors used crosslink mass spectrometry to characterize human Rix1 complex but these data are not discussed in context of the cryoEM structure of the subcomplex.

As mentioned above under the response for reviewer #1 we have re-written the crosslinking section to better emphasize what was learned about the complex and how this correlates with the cryo-EM structure of the subcomplex.

-The authors used the predicted alphafold models for the PELP1 Rix1 domain as an initial model for cryoEM refinement. The comparison of the final model of the complex with the alpha-fold model of the isolated protein should be discussed. This is important as the authors show that the LXXLL motifs are important for the stability of the domain and is unlikely, without conformational change, to be involved in nuclear receptor interaction. Can other region outside of the LXXLL motifs be involved in ER interaction?

Following the suggestion of the reviewer we have added a small discussion of the alpha-fold models. It is important to note that the alpha-fold model of human PELP1 is based on sequence and secondary structure similarity to previously determined structures of yeast Rix1. The deposited structures of yeast Rix1 are all in complex with Ipi3, the yeast homologue of WDR18, thus we don't have a good model for the structure of the isolated protein. In addition to the LXXLL motifs we think it is highly likely that the PXXP motifs, of which 2 of the 3 are disordered in the structure could potentially be involved in ER interaction indirectly by binding to Src.

-LXXLL motifs interact with nuclear receptors through non polar interactions but also formed electrostatic interactions through a charge clamp. What about sequence conservation of the various LXXLL motifs and

surrounding residues?

We thank the reviewer for the great suggestion to analyze the sequences surrounding each LxxLL motif. We aligned each of the LXXLL motifs from the Human PELP1 sequence with one another but did not observe any clear trends for additional conserved sequences reminiscent of a charged clamp. Interestingly several of the motifs including LM6-LM7 and LM10-LM11 are very close to one another.

```

                .....LxxLL.....
LM1  SAGSAAGVPGGTGGLSAVSSGPRLRLLLLESVSGLLQPRTGSAVAVPVHPPN
LM2  QPRTGSAVAVVHPPNRSAPHLPGIMCLLRLHGsvGGAQNLSALGALVSLSN
LM3  LGALVSLSNARLSSIKTRFEGLCLLSLLVGESPTLFFQQHCVSWLRSIQQV
LM4  WRSIQQVLQTDPPATMELAVAVLRDLLRYAAQLPALFRDISMNHLPGLIT
LM5  VLRDLLRYAAQLPALFRDISMNHLPGLITSLGLRPECEQSALEGMKACMT
LM6  SRLPSLGAGFSQGLKHTEWEQELHSLLASLHTLLGALYEGAETAPVQNEG
LM7  AGFSQGLKHTEWEQELHSLLASLHTLLGALYEGAETAPVQNEGPGVEMLL
LM8  ILDFICRTLSVSSKNISLHGDGPIRLLLLPSIHLEALDLSALILACGSRL
LM9  ELWVQVCGASAGMLQGGASGEALLTHLLSDISPPADALKLRSPRGSPDGSL
LM10 VMGVQQGEVLGSSPYTSSRCRRELYCLLLALLLPSPRCPPPLACALQAFSL
LM11 QGEVLGSSPYTSSRCRRELYCLLLALLLPSPRCPPPLACALQAFSLGQRED

```

Reviewer #4 (Remarks to the Author):

The manuscript authored by Gordon J and entitled "Architecture of the PELP1 molecular scaffold reveals insights into nuclear receptor coactivation" describes the purification and structural characterization of a core heterotetrameric Rix complex composed of two PELP1 and two WDR18 molecules. The cryo-EM structure shows that the eleven putative steroid hormone receptor binding motifs LxxLL present in each PELP1 molecule are either not solvent accessible or engaged in interactions that prevent SR binding. This finding provides a new mechanism for regulating SR interactions with the PELP1 co-activator. The structural results are sustained by in vivo coIP experiments showing that the Eostrogen receptor (ER) does not interact with the PELP1-WDR18 sub-complex. The structural results are of good quality but match an already published structure of the homologous yeast complex which in addition contains the IPI1/TEX10 subunit. The major new finding lays therefore in the positioning of the putative SR binding motifs which are absent in yeast.

The following issues need to be addressed:

1- P4: The initial description of the reconstituted RIX complex should mention the fact that PELP1 is strongly proteolyzed thus leaving only 20-30% of the intact full-length protein. This proportion is reduced when all 4 proteins are overexpressed but still leaves a mixture of 50% intact and 50% of cleaved PELP1. From the band intensities on the Coomassie stained gel the SENP3 and TEX10 subunits appear underrepresented. Therefore, the complex appears to be a mixture (described in the second section) and the authors cannot conclude at the end of section 1 that the complex is stoichiometric

We agree with the reviewer that the Coomassie stained gel shown in Fig. 1 does not represent a stoichiometric complex because of the degradation of the C-terminus of Pelp1 and we have removed this statement from the manuscript. To overcome the degradation issue, we moved the tag from the N-terminus of PELP1 to the C-terminus and found that this greatly reduced the amount of truncated PELP1 in our IPs. The Coomassie stained gel from the C-terminal tagged purified complex is shown in Supplementary Fig. 2 and we feel this represents a much more stoichiometric complex, however the exact stoichiometry of TEX10 and SENP3 remains unclear.

2- A C-terminal FLAG tagged PELP1 brings intact Rix1 complex which appears more relevant and homogeneous. The authors should indicate why was this complex not analysed in cryo-EM as it might have provided a more detailed view of the functional complex.

We completely agree that a structure of the full Rix1 complex would be great to have! We have prepared grids from the C-terminal FLAG tagged PELP1 complex, however we have yet to find the optimal

conditions for structure determination of the full complex. Thus far, the grids are dominated by the PELP1-WDR18 subcomplex, this could be because of instability of the complex at the air-water interface, or it could indicate that the N-terminal half of PELP1/WDR18 is connected to TEX10 and SENP3 through a very flexible tether, which will make it difficult if not impossible to visualize by cryo-EM.

3- Concerning the cross-linking mass spectrometry analysis, I was concerned about Supplemental figure 3 showing that only minute amounts of the complex is actually cross-linked. Does this affect the number and amounts of detectable cross-links? Secondly, I found that the cross-links between PELP1 and WDR16 are different in the Rix1 complex (supplemental fig3) than in the PELP1-WDR16 sub complex (supplemental 4). Additional cross-links are found for the PELP1-WDR16 sub complex as well as additional intralinks within PEPL1. This could mean that the structure of the subcomplex is different than that of the full Rix1 complex. The authors should comment these differences.

We have addressed most of these concerns above. The number of crosslinks detected is directly correlated with the amount of sample used. The PELP1/WDR18 subcomplex is very stable and we can isolate significantly higher quantities leading to the detection of more crosslinks, when compared to the full Rix1 complex. To make it easier to compare the crosslinks between the samples we have removed all crosslinks arising from epitope tags (which are not physiologically relevant). While we do not see as many crosslinks with the full complex, many of the same PELP1-WDR18 crosslinks are observed in both samples.

4- The title is misleading. This work addresses the structure of PEPL1 in a Rix1-sub complex which does not recapitulate the structure of free PELP1 which exerts the co-activator function. The structure of this very important Steroid Receptor co-activator can unfortunately not be determined (yet) in its functional state because of intrinsic flexibility without association with WRD18.

Following the suggestion of the reviewer we have changed the title of the manuscript to "Cryo-EM Reveals the Architecture of the PELP1-WDR18 Molecular Scaffold"

5- Based on the solvent accessibility and the availability of key Leucine residues, the authors conclude that the 11 LxxLL motifs are not compatible with steroid receptor binding. To support this conclusion, they provide Colp studies using an anti WRD18 antibody to show that ER α does not interact with WDR18 containing complexes (Fig.6a). However, control experiments shown in supplemental Fig 9 show that ER α interacts very weakly with PEPL1-containing complexes and the weak interaction can only be detected in a strongly overloaded lane. Therefore, only minute amounts of ER α are bound to free PELP which weakens the conclusions. I recommend that the authors perform binding experiments with purified steroid receptors.

We thank the reviewer for the excellent suggestion to perform binding experiments with purified steroid receptors. We purchased recombinant ERalpha (ThermoFisher, Cat # A15674) and carried out binding experiments with purified PELP1 in the presence and absence of E2, following a similar protocol to Yi et al. Molecular Cell, 2016. Unfortunately, we were not able to detect significant amounts of ERalpha binding to PELP1 under any of the conditions tested. While it is well established that PELP1 and ERalpha are interaction partners, the lack of binding between the recombinant proteins suggests that the interaction is not direct. This interaction might rely on additional factors such as p300 or Src, and/or it is dependent on some type of post-translational modification. Moreover, the interaction may be weak, and we could be working with concentrations of the proteins that are below the threshold for binding. Given that many additional variables could impact PELP1-ERalpha binding we chose not to include this negative result in the revised manuscript, but we have included the results below for the reviewer.

6- In the discussion the authors claim “Interestingly, we observed the N-terminal PELP1 LxxLL motifs LM1 and LM2 sampling ordered and disordered states upon 3D variability analysis of our cryo-EM data”. Unless I overlooked it, this data is not provided.

This was included as a supplemental video but it was not well referenced in the manuscript. We have updated the results to provide more details about the 3D variability experiments.

7- The authors provide an interesting discussion about the role of PELP1 overexpression in cancer cells which, to my opinion, included much speculation especially since the Rix1 complex has multiple additional roles in ribosome assembly and heterochromatin maintenance that could interfere with the oncogenic phenotype. It is also unknown what regulates the interactions between PELP1 and WDR18 is it only protein expression levels or can additional mechanisms such as post-translational modification be involved in addition.

Following the suggesting of the reviewer we have removed the majority of this discussion which we agree was speculative.

Minor comments

P5 PELP1 has a unique biochemical composition, It should read a unique amino acid composition

This has been corrected.

P6: The authors should mention in the main text that complex on which the XL-MS experiments were performed contains the C-terminally tagged version of PELP1.

We have added this comment to the main text.

P7 avoid repeats in the sentence "Single-particle reconstruction resulted in a 2.7 Å resolution reconstruction"

This sentence has been corrected.

Legend figure 2 based ON instead of based OFF

Thank you for catching this typo which has been corrected.

Figure 2 it is difficult to relate PELP1 structure in Fig 2f to the overall structure in Fig2e. there is a red colored helix in 2d which is not found in 2f, labeled.

We apologize for this confusion. The helix in 2d is from WDR18 and not PELP1. Upon reinspection of this figure we realized that the original colors used to indicate the PxxP motifs were too similar to the color used for WDR18 so it has been re-colored in the figure.

The authors should reference the supplemental figures 5-7 in the text

We have ensured that all supplemental figures are referenced in the text. We were missing a reference to Supplementary Fig. 7 so we thank the reviewer for catching this oversight.

Figure 4e, correct "blade contact with PELP1? – I'm not sure what they want us to correct the figure or the legend?"

Blade refers to the individual blades from the beta-propellers. We have re-phrased for clarity.

P12 mediated instead of mediated

This has been corrected.

REVIEWERS' COMMENTS

Reviewer #1 (Remarks to the Author):

This is a revised version of a previously submitted manuscript by Gordon et al. The manuscript has clearly improved and includes a revised model of the full Rix1 complex that is supported by additional experimental evidence. As per my main point of solving the structure of the entire Rix1 complex, the authors provide reasonable explanation for the technical challenges imposed by the inherent flexibility of the PELP1 C terminus in obtaining the full complex to properly behave in cryo-EM grids. Therefore, I agree that solving the structure of the full complex would be beyond the scope of the presented work. Although the structure of the full complex is not provided, the additional experiments included in the revision clearly demonstrate that PELP1 is unlikely a primary interaction partner of ERalpha. As previously stated, the information provided in this manuscript is critical to many fields as the Rix1 complex has emerged to be an important regulator of ribosome biogenesis and heterochromatin maintenance. In conclusion, the authors have addressed all my concerns in this revision, and I have no further comments.

Reviewer #2 (Remarks to the Author):

The authors have done an excellent job of responding the reviewer comments, particularly in response to questions about the lack of structural studies including full-length PELP1. There is some concern over the lack of estrogen mediated induction of endogenous genes in MCF7 cells and the overinterpretation of some of the results.

In regards to E2 induced genes; TFF1, PGR, GREB1, are genes that are typically robustly induced by E2. The luciferase based assays are not enough to make the conclusion that "WDR18 association with PELP1 prevents PELP1-mediated coactivation of ERa." Please note that mediated is spelled incorrectly on line 376.

Additionally, the authors have not demonstrated that WDR18 prevents the association between PELP1 and ERa. It is understood that the authors have not observed a robust interaction between ER and PELP1, but this is reported in the literature (as cited) by several other groups. Thus, it could be the IP conditions used by the authors. Additional data is needed to support this claim in the results section.

Reviewer #3 (Remarks to the Author):

The authors have addressed all my concerns and have significantly improved the quality of the manuscript.

Reviewer #4 (Remarks to the Author):

The authors have thoroughly revised their manuscript by performing additional experiments, including suggestions and corrections. They addressed point by point all the comments I raised and I hope that the quality of the manuscript is improved. The current version is ready to be published in Nature Communications.

Rebuttal NCOMMS-22-17303

Reviewer Comments (*black italics*) and our response (*blue*).

Reviewer #1 (Remarks to the Author):

This is a revised version of a previously submitted manuscript by Gordon et al. The manuscript has clearly improved and includes a revised model of the full Rix1 complex that is supported by additional experimental evidence. As per my main point of solving the structure of the entire Rix1 complex, the authors provide reasonable explanation for the technical challenges imposed by the inherent flexibility of the PELP1 C terminus in obtaining the full complex to properly behave in cryo-EM grids. Therefore, I agree that solving the structure of the full complex would be beyond the scope of the presented work. Although the structure of the full complex is not provided, the additional experiments included in the revision clearly demonstrate that PELP1 is unlikely a primary interaction partner of ERalpha. As previously stated, the information provided in this manuscript is critical to many fields as the Rix1 complex has emerged to be an important regulator of ribosome biogenesis and heterochromatin maintenance. In conclusion, the authors have addressed all my concerns in this revision, and I have no further comments.

We thank the reviewer for their support of our revised manuscript!

Reviewer #2 (Remarks to the Author):

The authors have done an excellent job of responding the reviewer comments, particularly in response to questions about the lack of structural studies including full-length PELP1.

We thank the reviewer for their supportive comments.

There is some concern over the lack of estrogen mediated induction of endogenous genes in MCF7 cells and the overinterpretation of some of the results. In regards to E2 induced genes; TFF1, PGR, GREB1, are genes that are typically robustly induced by E2. The luciferase based assays are not enough to make the conclusion that "WDR18 association with PELP1 prevents PELP1-mediated coactivation of ERa." Please note that mediated is spelled incorrectly on line 376.

We completely agree with the reviewer that additional work is needed to fully establish if WDR18 association with PELP1 prevents PELP1-mediated coactivation of ERalpha. TFF1, PGR, and GREB1 are well established E2 genes, however we are not aware of any experiments directly showing that these E2 genes are influenced by PELP1. To fully test this hypothesis, we need a group of well characterized E2 genes that are activated by PELP1. In our first revision we selected a handful of PELP1 targets that had been identified following PELP1 si-RNA knockdown in ZR75 cells (Mann et al Molecular Oncology, 2013; <https://doi.org/10.1016/j.molonc.2013.12.012>), unfortunately we did not detect a robust estrogen response with these genes in MCF7 cells. This result may indicate that these genes are cell-type specific. The other possibility is that ERalpha activating complexes are very heterogenous and PELP1 is only impacting a small percentage making it very challenging to find and identify testable targets.

To address the concerns of the reviewer we have modified the discussion to emphasize that additional studies are needed to understand PELP1 coactivation of ERalpha. We hope that our structural work will aid in the design of new experiments to test the working hypothesis that WDR18 association with PELP1 prevents ERalpha coactivation. Given that PELP1 has also

been shown to co-activate many other nuclear receptors we hope that our work will be instrumental in the development of cell-based approaches to see how WDR18 association with PELP1 influences the coactivation of other nuclear receptors.

Additionally, the authors have not demonstrated that WDR18 prevents the association between PELP1 and ERα. It is understood that the authors have not observed a robust interaction between ER and PELP1, but this is reported in the literature (as cited) by several other groups. Thus, it could be the IP conditions used by the authors. Additional data is needed to support this claim in the results section.

We have tried numerous approaches (cell based and with recombinant proteins) but failed to observe a robust interaction with ERα and PELP1. There are several published IP experiments showing an interaction between ERα and PELP1. As suggested by the reviewer our inability to reproduce these experiments could be the result of differences in experimental setup (cell line, antibody, hormone exposure, placement of epitope tag, etc) or it may indicate that the association between ERα and PELP1 is transient and/or indirect. We favor the hypothesis that the PELP1-ERα interaction is indirect given the large number of interaction partners for PELP1 that have been identified in the literature to date.

To address the concerns of the review we have carefully modified the results and discussion to emphasize that more studies are needed to determine if WDR18 association with PELP1 prevents association with ERα.

Reviewer #3 (Remarks to the Author):

The authors have addressed all my concerns and have significantly improved the quality of the manuscript.

Thank you!

Reviewer #4 (Remarks to the Author):

The authors have thoroughly revised their manuscript by performing additional experiments, including suggestions and corrections. They addressed point by point all the comments I raised and I hope that the quality of the manuscript is improved. The current version is ready to be published in Nature Communications.

We thank the reviewer for their supportive comments!